# Trehalose increases tomato drought tolerance, induces defenses, and increases resistance to bacterial wilt disease

April M. MacIntyre[1,2¤a], Valerian Meline[3], Zachary Gorman[4], Steven P. Augustine[5], Carolyn J. Dye[1¤b], Corri D. Hamilton[1], Anjali S. Iyer-Pascuzzi[3], Michael V. Kolomiets[4], Katherine A. McCulloh[5], Caitilyn Allen[1]*

1 Department of Plant Pathology, University of Wisconsin-Madison, Madison, WI, United States of America,
2 Department of Bacteriology, University of Wisconsin-Madison, Madison, WI, United States of America,
3 Department of Botany and Plant Pathology, Purdue University, West Lafayette, IN, United States of America, 4 Department of Plant Pathology and Microbiology, Texas A&M University, College Station, TX, United States of America, 5 Department of Botany, University of Wisconsin-Madison, Madison, WI, United States of America

¤a Current address: Valent Biosciences, Libertyville, IL, United States of America
¤b Current address: Soil Carbon Co., Minneapolis-St. Paul, MN, United States of America
* callen@wisc.edu

**Data Availability Statement:** All relevant data are within the manuscript and its Supporting Information files.

## Abstract

*Ralstonia solanacearum* causes bacterial wilt disease, leading to severe crop losses. Xylem sap from *R. solanacearum*-infected tomato is enriched in the disaccharide trehalose. Water-stressed plants also accumulate trehalose, which increases drought tolerance via abscisic acid (ABA) signaling. Because *R. solanacearum*-infected plants suffer reduced water flow, we hypothesized that bacterial wilt physiologically mimics drought stress, which trehalose could mitigate. We found that *R. solanacearum*-infected plants differentially expressed drought-associated genes, including those involved in ABA and trehalose metabolism, and had more ABA in xylem sap. Consistent with this, treating tomato roots with ABA reduced both stomatal conductance and stem colonization by *R. solanacearum*. Treating roots with trehalose increased xylem sap ABA and reduced plant water use by lowering stomatal conductance and temporarily improving water use efficiency. Trehalose treatment also upregulated expression of salicylic acid (SA)-dependent tomato defense genes; increased xylem sap levels of SA and other antimicrobial compounds; and increased bacterial wilt resistance of SA-insensitive *NahG* tomato plants. Additionally, trehalose treatment increased xylem concentrations of jasmonic acid and related oxylipins. Finally, trehalose-treated plants were substantially more resistant to bacterial wilt disease. Together, these data show that exogenous trehalose reduced both water stress and bacterial wilt disease and triggered systemic disease resistance, possibly through a Damage Associated Molecular Pattern (DAMP) response pathway. This suite of responses revealed unexpected linkages between plant responses to biotic and abiotic stress and suggested that *R. solanacearum*-infected plants increase trehalose to improve water use efficiency and increase wilt disease resistance. The pathogen may degrade trehalose to counter these efforts. Together, these results suggest that treating tomatoes with exogenous trehalose could be a practical strategy for bacterial wilt management.

**Funding:** This research was supported by a USDA-NIFA Predoctoral Fellowship to AMM and by the University of Wisconsin-Madison College of Agricultural and Life Sciences. The funders had no role in study design, data collection and analysis, decision to publish, or preparation of the manuscript.

**Competing interests:** The authors have declared that no competing interests exist.

## Introduction

*Ralstonia solanacearum* is a vascular plant pathogen that causes bacterial wilt disease in over 200 species of plants worldwide [1]. It is widely studied because it is economically damaging to food, fiber, and feed production [2, 3]. *R. solanacearum* is also notoriously difficult to eradicate because it persists long-term in soil, water, and latently infected hosts [4].

Most *R. solanacearum* strains infect plants through the soil [3]. *R. solanacearum* chemo- or aero-taxes towards roots, evades root border cell traps, enters roots at points of lateral root emergence, moves to the root cortex, and eventually migrates to the xylem, all with the help of secreted proteins such as hydrolytic enzymes and Type III secreted effectors [5–12]. Once in xylem vessels, *R. solanacearum* forms biofilm matrices composed of extracellular DNA and polysaccharide (EPS) and proliferates to $>10^9$ CFU/g of stem tissue [13–15]. *R. solanacearum* also conditions the xylem environment to favor bacterial growth [16]. The large amounts of bacteria and EPS clog xylem vessels and disrupt water flow [17]. If the plant cannot contain bacterial growth and spread, it will wilt and eventually die.

Researchers long assumed that *R. solanacearum* and other vascular pathogens cause drought-like physiological symptoms in host plants and that severity of the disease is affected by environmental water availability [18–20]. *R. solanacearum* dies more quickly in drought-stressed chickpea plants, and *R. solanacearum* infection and water stress induce similar osmolyte biosynthesis and expression of genes related to the plant signaling molecule abscisic acid (ABA), a well-characterized marker of water stress [21, 22]. Further, stomatal conductance decreases in *R. solanacearum* infected plants [16]. These studies suggest but do not prove that plant response to bacterial wilt mimics the drought stress response.

During water and osmotic stress, many plants, including tomatoes, accumulate trehalose and trehalose-6-phosphate (T6P), which contribute to tolerance of these stresses [23–27]. Xylem sap from *R. solanacearum*-infected tomato plants contains about 20-fold more trehalose than healthy plants [16]. This non-reducing disaccharide composed of glucose and UDP-glucose is best known for its role as a compatible solute. Trehalose synthesis helps *R. solanacearum* tolerate osmotic stress and contributes to pathogen fitness and virulence [28]. However, the tomato host produces the trehalose in xylem sap during infection; *R. solanacearum* does not export its trehalose in culture and it aggressively expresses the secreted trehalase TreA *in planta* [29].

The roles of trehalose and T6P as nutritional signaling metabolites in plants have been reviewed [30–34]. In *Arabidopsis*, most trehalose metabolism-related proteins have predicted functions in development or signaling [35, 36]. The current paradigm is that T6P, and by extension trehalose, are direct indicators of sucrose levels that help plants maintain sucrose-to-starch ratios through SnRK1 signaling [37–40]. Trehalose also likely improves drought tolerance in plants by two mechanisms: by regulating carbohydrate levels during stress conditions, possibly via the SnRK1 kinase; and by altering ABA signaling and stomatal conductance [23, 41, 42]. Members of another kinase family, SnRK2, also integrate with ABA signaling, and trehalose metabolism may connect to this network via the TOR regulator [43, 44].

Some studies suggest that trehalose may mediate host-microbe interactions and contribute to plant defense. For example, treating *Arabidopsis* seedlings in liquid culture with 30 mM exogenous trehalose triggered the expression of defense genes in ethylene and methyl-jasmonate pathways [45]. Infusing tobacco leaves with a 10–50 mM trehalose solution increased resistance to tobacco mosaic virus and triggered a transcriptomic response suggestive of biotic and abiotic stress tolerance [46]. A *Xanthomonas citri* trehalose synthesis mutant had reduced fitness and infusing citrus leaves with a trehalose solution induced defense gene expression, suggesting that trehalose might simultaneously be a virulence factor and an indicator of

pathogen attack [47]. In another study, spraying tomato leaves with the trehalase inhibitor Validamycin A increased resistance to bacterial wilt and *Fusarium* wilt and activated systemic acquired resistance against a wide range of other pathogens in *Arabidopsis* [48–50]. Additionally, silencing some tomato trehalose-6-phosphate synthase genes decreased resistance to *Botrytis cinerea* and *P. syringae* pv. *tomato* DC3000 and affected expression of JA, ET, and SA signaling responsive, defense related genes [51].

We investigated the relationship between bacterial wilt and drought stress in tomatoes and explored the role of trehalose in this interaction. We tested the following hypotheses: 1) *R. solanacearum infection mimics abiotic stress triggered by disrupted water flow;* and 2) *Exogenous trehalose inhibits bacterial wilt development in tomatoes by enhancing plant water use efficiency and inducing defense response*s. To test $H_o1$, we used meta-transcriptomic analyses to show that tomato genes involved in trehalose metabolism and water stress responses are differentially expressed during *R. solanacearum* infection in susceptible tomato hosts. Using ABA treatment to close stomates, we found that *R. solanacearum* depends on stomatal conductance and water flow for effective host colonization. We also discovered that *R. solanacearum* infection increases xylem sap ABA levels.

Testing $H_o2$ revealed that treating plants with exogenous trehalose improved water use efficiency in drought-stressed tomatoes, decreased short-term stomatal conductance and transpiration, and increased ABA concentration in xylem sap. Trehalose-treated tomato plants were more resistant to bacterial wilt. This protective effect appeared to be mediated by up-regulation of salicylic acid (SA)- and jasmonic acid (JA)-related defense genes and by increased levels of jasmonates and other defense compounds in xylem sap.

## Results

### Bacterial wilt disease altered expression of tomato trehalose and water stress genes

To investigate the relationship between bacterial wilt and drought stress, we measured expression of all tomato genes annotated as related to trehalose metabolism, ABA, or water stress responses. Candidate genes included those known or predicted to encode synthesis and degradation of trehalose, including trehalose-6-phosphate phosphatase (TPP), trehalose-6-phosphate synthase (TPS), and trehalase [35, 36, 52]. Genes encoding water stress-associated proteins included: 1) those involved in biosynthesis, regulation, and response to ABA, a known indicator of water stress [53, 54]; 2) late embryogenesis/dehydrins (LEAs), which accumulate in water-stressed plants [55]; and 3) aquaporins, channel proteins that allow living cells to move water and small solutes more efficiently [56].

We used four datasets that profiled transcriptomes of two wilt-susceptible tomato plants (WV700 and Bonny Best) responding to infection by *R. solanacearum* strain K60 or GMI1000 [57]. After normalizing the datasets to allow comparisons across these experiments, we measured the expression of these candidate genes in healthy plants relative to their levels in *R. solanacearum*-infected plants.

The datasets came from diverse experiments involving different plant genotypes, tissues, and *R. solanacearum* strains. Despite considerable experiment-associated variation across datasets, this analysis revealed that *R. solanacearum*-infected tomato plants significantly changed the expression of several water stress physiology genes (Fig 1). Nine genes meeting the annotation criteria were significantly differentially expressed across all datasets. Notably, the trehalose-6-phosphate synthase gene *SlTPS7* was upregulated in all of the susceptible tomatoes in response to *Rs*, suggesting this may be the main T6P protein responsible for elevated trehalose levels in xylem sap during infection. All infected plants upregulated an ABA

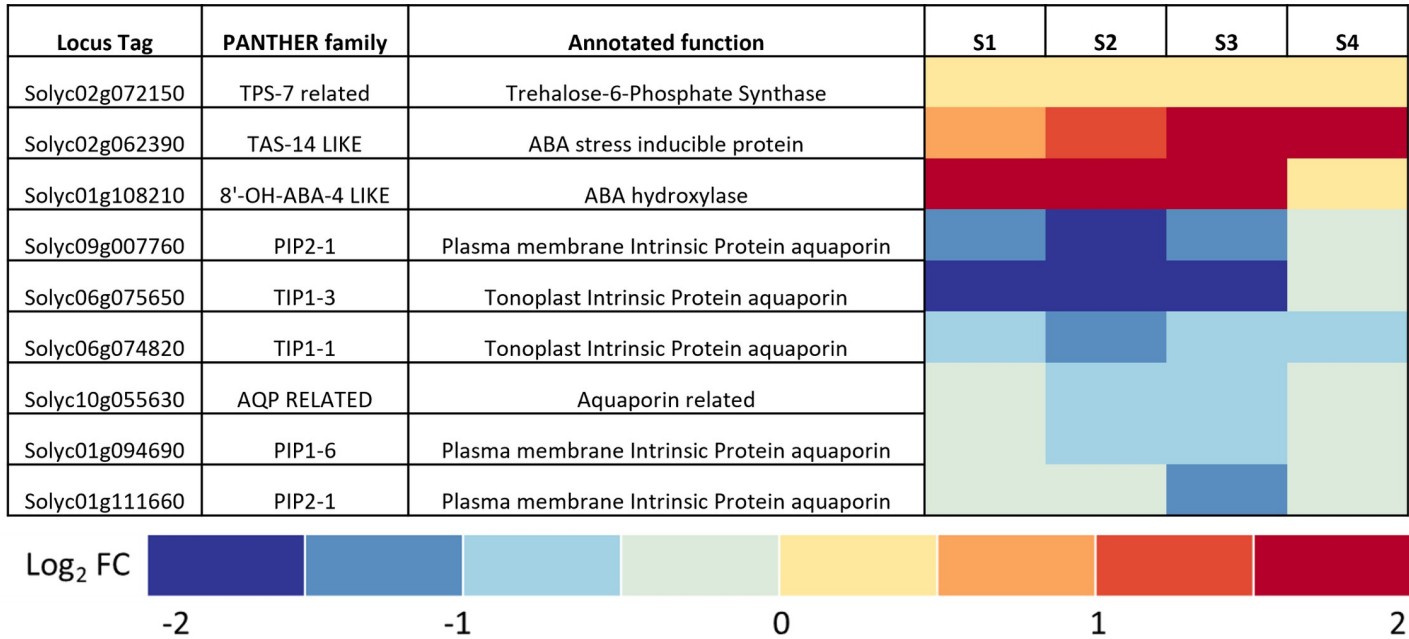

| Locus Tag | PANTHER family | Annotated function | S1 | S2 | S3 | S4 |
|---|---|---|---|---|---|---|
| Solyc02g072150 | TPS-7 related | Trehalose-6-Phosphate Synthase | | | | |
| Solyc02g062390 | TAS-14 LIKE | ABA stress inducible protein | | | | |
| Solyc01g108210 | 8'-OH-ABA-4 LIKE | ABA hydroxylase | | | | |
| Solyc09g007760 | PIP2-1 | Plasma membrane Intrinsic Protein aquaporin | | | | |
| Solyc06g075650 | TIP1-3 | Tonoplast Intrinsic Protein aquaporin | | | | |
| Solyc06g074820 | TIP1-1 | Tonoplast Intrinsic Protein aquaporin | | | | |
| Solyc10g055630 | AQP RELATED | Aquaporin related | | | | |
| Solyc01g094690 | PIP1-6 | Plasma membrane Intrinsic Protein aquaporin | | | | |
| Solyc01g111660 | PIP2-1 | Plasma membrane Intrinsic Protein aquaporin | | | | |

$Log_2$ FC: -2, -1, 0, 1, 2

**Fig 1. *R. solanacearum* infection consistently changes expression of nine tomato genes involved in trehalose metabolism and water stress physiology.** Meta-analysis of transcriptomic data from four different experiments measuring gene expression of tomato plants responding to infection by *R. solanacearum* identified nine genes of interest that were commonly differentially expressed across all data sets. The heatmaps show relative expression (as $log_2$ fold change) compared to mock-inoculated control plants of all tomato genes that were both: 1) Annotated with the terms *trehalose*, *drought*, or *abscisic acid / ABA*; and 2) Identified by the meta-transcriptomic analysis as significantly differentially expressed across four different tomato RNAseq studies. The four studies were: **S1**, roots of wilt-susceptible West Virginia 700 tomato plants (WV) sampled 24 h after infection with *R. solanacearum* strain K60; **S2**, roots of WV sampled 48 h after infection with *R. solanacearum* strain K60; **S3**, seedling roots of wilt-susceptible cv. Bonny Best (BB) sampled 24 h after infection with *R. solanacearum* strain GMI1000; **S4**, mid-stems of BB sampled 72 h after inoculation with *R. solanacearum* strain GMI1000. Tomato locus tags, gene names, and PANTHER families are from NCBI and the Sol Genomics Network, accessed in 03/2020. Transcriptomic data were normalized across experiments to allow comparisons as described in Methods.

hydroxylase and an ABA- and environmental stress-inducible dehydrin protein TAS14 and downregulated many diverse aquaporins, consistent with transcript changes observed in plants undergoing abiotic water stress. These results led us to directly measure physiological and metabolic indicators of drought stress in *R. solanacearum*-infected plants.

## Bacterial wilt disease increased ABA concentration in xylem sap

Plants can use ABA to mitigate drought stress. This plant hormone, synthesized locally in guard cells or the vasculature, closes stomates to reduce water loss through transpiration [58]. To test the hypothesis that tomato plants produce ABA in response to bacterial wilt, we used LCMS/MS to compare ABA levels in xylem sap of healthy plants and *R. solanacearum*-inoculated plants showing early wilt symptoms. Xylem sap from the *R. solanacearum*-infected plants contained 40% more ABA than sap from mock-inoculated plants (Fig 2A).

## *R. solanacearum* depends on host stomatal conductance to establish infection

We previously observed that *R. solanacearum*-infected tomatoes had reduced stomatal conductance [59]. To test the hypothesis that the increased ABA levels and lowered stomatal conductance in infected plants also affect bacterial wilt disease development, we measured the effects of exogenous ABA treatment on stomatal conductance, transpiration rates, and *R. solanacearum* colonization of susceptible tomato plant stems. We watered tomato plants daily

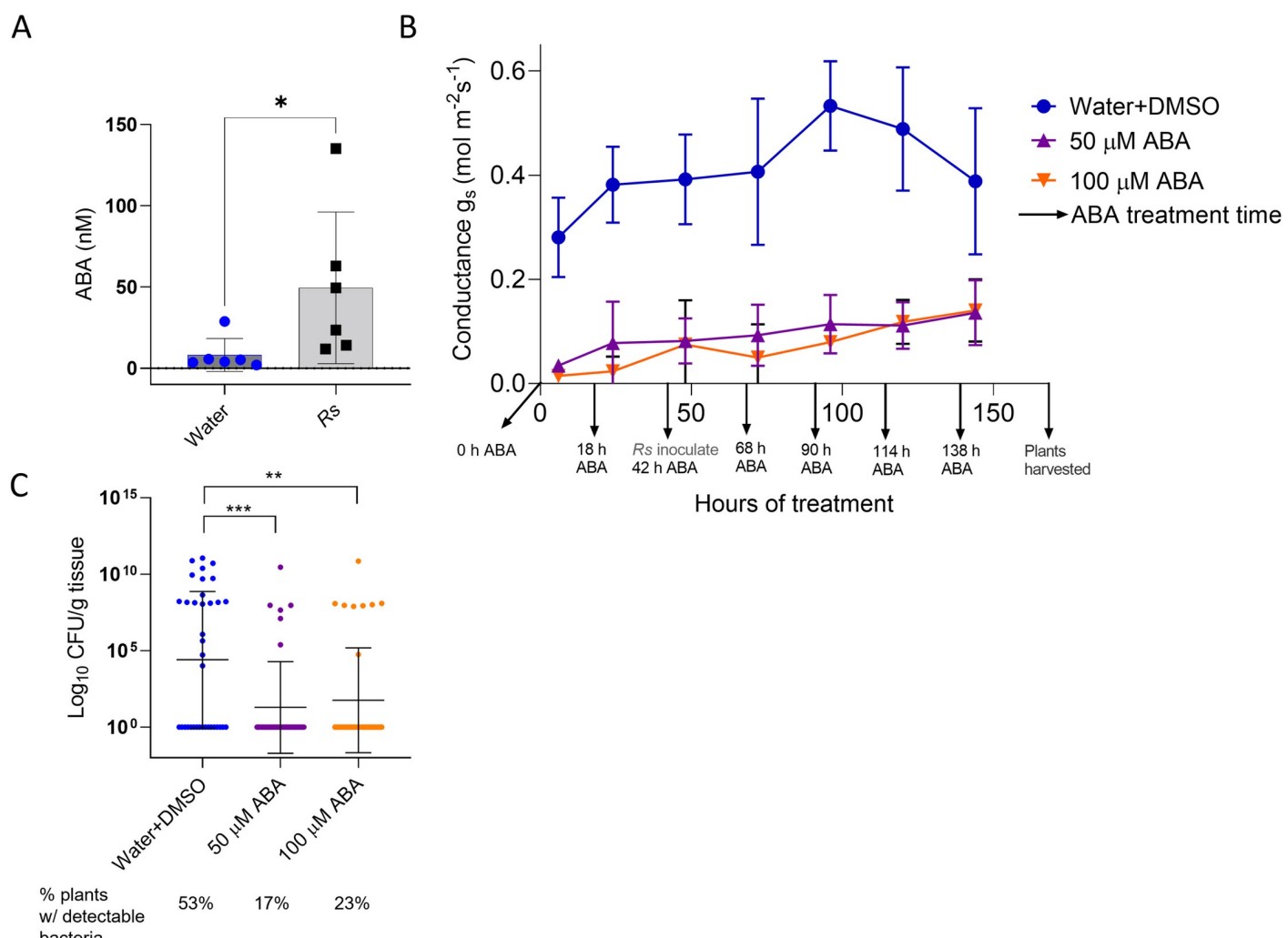

**Fig 2. ABA concentration increased in *R. solanacearum*-infected plants, and ABA treatment reduced stomatal conductance, which correlated with reduced *R. solanacearum* stem colonization. A)** ABA concentration in xylem sap harvested from healthy and *R. solanacearum*-infected Bonny Best tomato plants showing early wilt symptoms (1–25% of leaf area wilted) (T-test, $^*P$ = .03). Bars show mean and standard deviation of xylem sap ABA levels, measured by GC-MS, from six plants per treatment. **B)** Stomatal conductance in tomato leaves as measured with LI-COR instrument. Conductance of plants watered with 50 or 100 uM of ABA differed from that of mock-inoculated controls at all time points (2-way ANOVA Tukey's multiple comparisons, $P < .05$). Data represent four biological replicate experiments, each with three plants analyzed per treatment per time point. Bars represent standard deviation. **C)** *R. solanacearum* population sizes as quantified in tomato plant stems five days after soil-soak inoculation with *R. solanacearum* strain GMI1000. *R. solanacearum* colonization was lower in ABA-treated plants at 5 dpi, which corresponds to 168 h of ABA treatment (Kruskal Wallis, 50 uM ABA, $^{***}P < .001$; 100 uM ABA, $^{**}P < .01$). The proportion of ABA-treated plants containing detectable *R. solanacearum* cells (shown below X-axis labels) was also lower (Chi-square, $P$ = .0016). Dots represent *R. solanacearum* population sizes in individual plants. Black bars represent the geometric mean and standard deviation of the data. Plants with undetectable *R. solanacearum* populations were given a value of 1 for statistical analysis. Data represent three biological replicate experiments, each with 12 plants per treatment (36 total plants/treatment).

with 50 mL of either 100 μM ABA, 50 μM ABA, or water as control and measured stomatal conductance four to six hours later. After 48 h of ABA treatment, plants were drench-inoculated by adding a suspension of *R. solanacearum* strain GMI1000 to the ABA treatment solutions, which we determined were neither nutritious nor cytotoxic to *R. solanacearum* (S1 Fig). Treatment with ABA reliably reduced stomatal conductance in the plants over the seven days of the experiment; 100 μM and 50 μM ABA treatment had similar effects (Fig 2B).

ABA treatments continued for four days after plants were inoculated with *R. solanacearum*. At five dpi we measured *R. solanacearum* population sizes in the stems. Stems of ABA-treated plants were less frequently colonized by *R. solanacearum* than control plants (50 μM ABA = 17%

plants colonized; 100 μM = 23% plants colonized; water control = 53% plants colonized) and they contained lower *R. solanacearum* population sizes than untreated plants (Fig 2C). The correlation between ABA treatment, reduced stomatal conductance, and lower *R. solanacearum* colonization in tomato stems suggests that plant water movement assists *R. solanacearum* infection either at the roots or in the stem and that reducing host water flow reduces or slows *R. solanacearum* spread in the plant.

## Trehalose treatment transiently improved water use efficiency in tomatoes

Tomato plants become more drought tolerant following exposure to exogenous trehalose via ABA signaling and stomatal closure [42]. Because ABA and trehalose concentrations increased in xylem sap from *R. solanacearum*-infected plants and pathogen spread depends on host xylem water transport, we asked whether exogenous trehalose could also protect plants from bacterial wilt.

We first confirmed that trehalose increases tomato drought tolerance under our experimental conditions. Two-week-old wilt-susceptible Bonny Best tomato plants were treated with 50 mL of 30 mM trehalose solution or water and then were not watered for the rest of the experiment. Trehalose-treated plants remained turgid for three days after the water-treated control plants began wilting, showing that trehalose treatment delayed wilting from lack of water under our conditions (Fig 3A). Quantitative mass spectrometry analysis of tomato xylem sap revealed that ABA content was 5-fold higher in trehalose-treated plants at 24 h (Fig 3B). However, at 6 and 48 h after treatment, ABA levels were no different from those of water-treated control plants. Gravimetric assays quantifying water loss in trehalose-treated plants compared to water controls confirmed that trehalose-treated plants lose water at a slower rate due to transpiration than controls (data not shown).

Next, we assessed stomatal conductance, transpiration, and photosynthesis of trehalose-treated plants. We measured tomato leaf responses with a LI-COR portable photosynthesis machine at 10 min, 6, 24, 48, and 72 h after treatment with 30mM trehalose or water. Transpiration in trehalose-treated plants was lower than in controls at 6, 24, and 48 h (Fig 3C). Relative to untreated plants, trehalose treatment detectably reduced stomatal conductance after just 10 min, an effect that persisted for at least 6 h. By 24 h after treatment, stomatal conductance of treated plants was indistinguishable from that of water controls (Fig 3D). Water use efficiency, the relationship between water loss and plant productivity (photosynthesis/transpiration), was temporarily improved at 6 h by trehalose treatment (S2A Fig). Trehalose treatment did not negatively affect photosynthesis (S2B Fig).

We found that tomato plant growth was not significantly altered by trehalose treatment and the leaf relative water content was the same in trehalose-treated and water-treated plants at 48h (S2C Fig). Further, 1–3 days of watering with trehalose solution did not affect tomato seedling root growth *in vitro* relative to water controls (S2D Fig). Chlorophyll content, measured by SPAD meter, was transiently lower in trehalose treated plants 2–10 days after treatment, although it eventually returned to the same levels as in untreated plants (S2E Fig). Together, these diverse physiological and biochemical analyses showed that treating tomato plants with exogenous trehalose temporarily increased their drought tolerance, likely because trehalose increased ABA levels and ABA signaling closes stomata.

## Trehalose is a potent inhibitor of bacterial wilt disease

After finding that bacterial wilt impacts host water stress physiology and that exogenous trehalose reduces stomatal conductance in tomatoes, we wondered if the decreased water movement in trehalose-treated plants would slow bacterial wilt disease development. We therefore

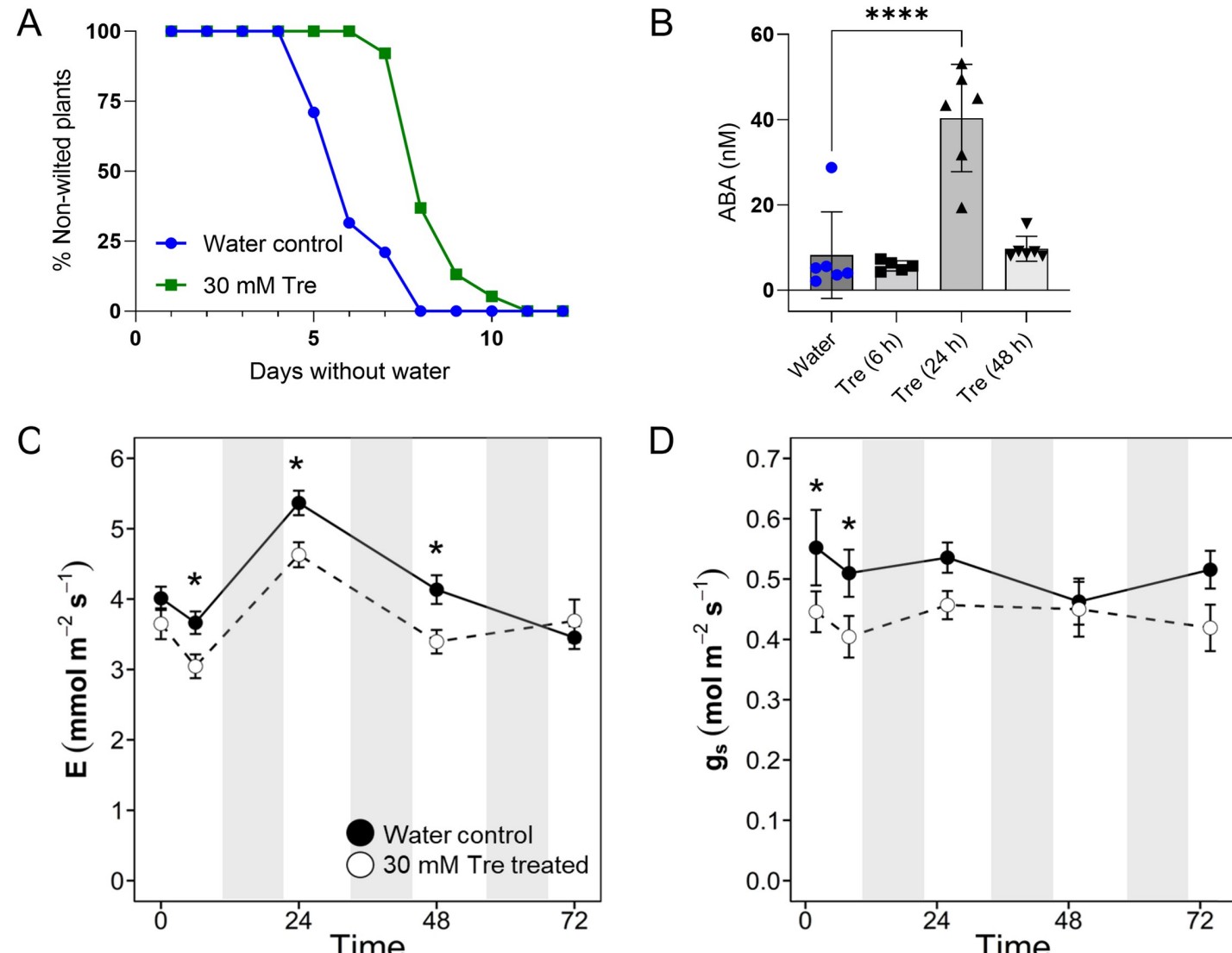

**Fig 3. Trehalose treatment delayed tomato wilting due to drought, likely by transiently increasing ABA content in xylem sap and by decreasing stomatal conductance and transpiration. A)** Curves show how long it took plants to wilt following a single treatment with either water (blue) or 30 mM trehalose (green) and then no water for two weeks ($P$ = .01, T-test of areas under the curve). Data shown represent four biological replicates totaling 38 plants per treatment. **B)** ABA concentration in trehalose treated plants (one-way ANOVA Fisher's LSD multiple comparisons to water-treated control: 6 h tre $P$ = .63; 24 h tre $P$ < .0001; 48 h tre $P$ = .76). Water controls were collected concurrently with the treated samples and averaged. Data shown represent xylem sap samples from six plants. **C)** Transpiration and **D)** stomatal conductance were measured with LI-COR. Asterisks indicate $P$>.05, mixed-model ANOVA, Tukey's HSD. Data shown represent 20 plants/treatment. Measurements were taken 10 min after treatment with trehalose or water, then each day after that in the morning two hours after growth chamber lights went on. Shaded gray vertical bars on the graphs represent night periods. Bars on the data points represent the standard deviation. All data were calculated from the same LICOR time points.

tested the hypothesis that trehalose makes plants more wilt resistant. Forty-eight hours after plants had been treated with 30 mM trehalose solution or water, they were drench inoculated with a suspension of *R. solanacearum* bacteria. Bacterial wilt incidence and severity was reduced by 25–75% across biological replicates in the trehalose-treated plants, demonstrating that exogenous trehalose increases tomato resistance to bacterial wilt (Fig 4A).

Since *R. solanacearum* can infect plants without causing visible symptoms, the absence of wilt in trehalose treated plants could indicate either the lack of disease or latent infection. To

distinguish between these possibilities, we measured *R. solanacearum* colonization of tomato midstems at 5, 6, and 7 dpi. Water-treated control plants were three times more likely to contain detectable *R. solanacearum* cells than trehalose-treated plants (20.7% compared to 7.4%), and control plants contained significantly higher bacterial population sizes (Fig 4B). This showed that trehalose treatment did not increase the frequency of latent infections but rather that *R. solanacearum* was less likely to colonize stems of trehalose treated plants. The reduced colonization frequency at this relatively early time point is consistent with the observed trehalose-associated delay in disease development.

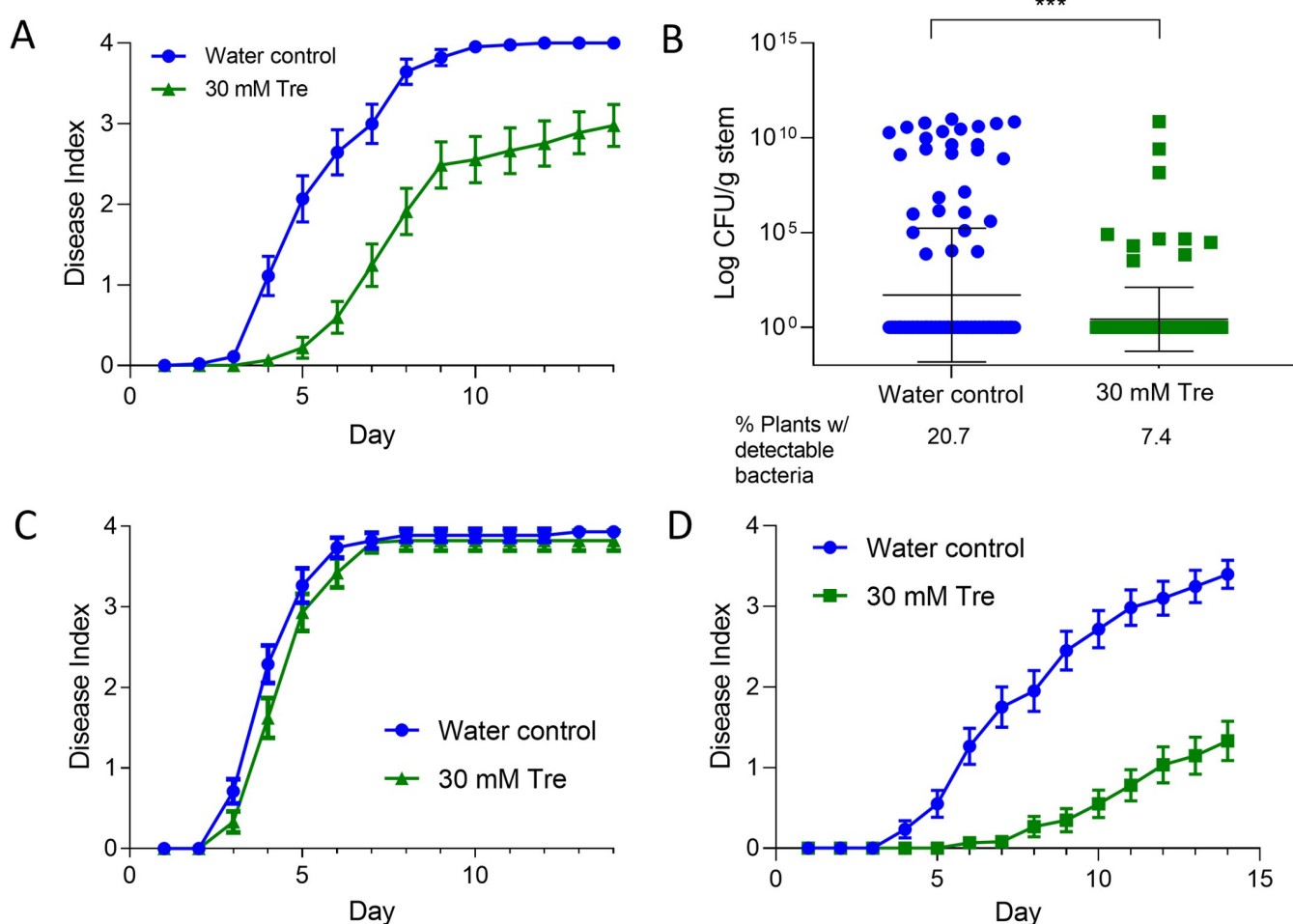

**Fig 4. Treating plants with trehalose at the root delayed bacterial wilt symptoms and reduced *R. solanacearum* stem colonization, but only following soil inoculation. A)** Wilt disease progress of cv. Bonny Best tomato plants treated with trehalose, then after 48 h soil soak inoculated with *R. solanacearum* GMI1000. Disease symptoms were rated on a 0–4 disease index scale where 0 = no wilt, and 4 = 76–100% wilted or dead. Trehalose treatment delayed appearance of wilt symptoms ($P$ = .0082, t-test comparing areas under the curves or AUC). Data represent three replicate experiments, each with 15 plants/ treatment. Bars represent standard error. **B)** Bacterial population sizes were smaller in stems of trehalose-treated plants at 5–7 days after inoculation (Mann-Whitney, $P$ = .0009). Data represent three biological replicates with 15 plants/day sampled/treatment; each dot represents the *R. solanacearum* population size in one plant (including plants without detectable *R. solanacearum*). Black bars represent geometric mean and standard deviation of the data. The percentage of plants that contained any detectable *R. solanacearum* cells is shown below the X-axis; the limit of detection is ~$10^2$ cells. Relative to controls, fewer trehalose-treated plants contained detectable *R. solanacearum* populations (Chi-square, $P$ = .0016). **C)** Disease progress of plants treated with trehalose (green) or water (blue), then 48 h later inoculated by introducing *R. solanacearum* directly into the xylem through a cut leaf petiole. Trehalose treatment did not affect wilt disease development in petiole-inoculated plants ($P$ = .2067, T-test of AUC). Data represent three replicate experiments, each containing 15 plants/treatment. Bars represent standard error. **D)** Disease progress of plants treated with 30 mM trehalose or water, then 120 h later inoculated with *R. solanacearum* the soil drench method ($P$ = .0003, T-test of AUC) Data represent four replicate experiments, each containing 15 plants/treatment. Bars represent standard error of the mean.

We tested the protective limits of trehalose treatment by varying inoculation sites, treatment methods, and pathogens. Interestingly, inoculating *R. solanacearum* cells directly into the xylem through a cut petiole overwhelmed any protective benefit of trehalose treatment (Fig 4C). This result suggests that *R. solanacearum* must overcome a trehalose-enhanced checkpoint at or in tomato roots to colonize plant stems. However, the protective effect of root drenching is enduring; trehalose treatment reduced bacterial wilt incidence even when plants were inoculated five days later (Fig 4D).

*R. solanacearum* produces a trehalase enzyme and can use trehalose as a carbon source, so we wondered if treating tomato roots with trehalose nutritionally enriched the soil environment, thereby de-incentivizing *R. solanacearum* root invasion. We tested this hypothesis by inoculating trehalose-treated tomato plants with either wild-type *R. solanacearum* or a Δ*treA* trehalase mutant that cannot grow on trehalose [28]. Trehalose treatment similarly reduced disease caused by wild type and Δ*treA R. solanacearum*, so bacterial catabolism of trehalose cannot explain trehalose treatment's protective effect (S3A Fig).

Spraying trehalose solution onto leaves and then inoculating *R. solanacearum* by soil-soak 48 h later also did not affect bacterial wilt progress (S3B Fig). To explore the effects of trehalose treatment on another bacterial pathogen, we dip-inoculated tomato leaves with the tomato leaf spot pathogen *Xanthomonas gardneri* 72 h after plant roots were treated with trehalose. Trehalose had no significant effect on leafspot disease severity or *X. gardneri* colonization of leaves sampled 3, 6, and 9 dpi (S3C Fig). These experiments demonstrated that trehalose needs to be applied at the roots to protect against bacterial wilt and that trehalose might increase resistance only to *R. solanacearum*.

## Exogenous trehalose increases the expression of tomato defense genes, and SA signaling may mediate its protective effect

Although trehalose treatment of roots decreased plant transpiration and stomatal conductance temporarily (6–48 h), this treatment protected plants from bacterial wilt for at least five days (Fig 4D), suggesting that trehalose induced plant defense responses. We tested this by quantifying the effect of trehalose treatment on defense gene expression, bacterial wilt resistance of defense hormone-insensitive tomato plants, and levels of hormones and defense compounds.

To measure the effect of trehalose on defense gene expression, we extracted RNA from tomato stem sections 6 or 48 h after roots were treated with 30 mM trehalose solution or water. We measured expression of two JA-responsive genes, *PIN2* and *LOXA* [60, 61]. *PIN2* was significantly downregulated 6 h after trehalose treatment (Fig 5A), suggesting JA levels may have been affected by trehalose treatment, while *LOXA* expression did not change after trehalose treatment. *GLUA*, *PR1A*, and *PR1B* are tomato pathogenesis-related (PR) genes induced by SA and by pathogen challenge [62–64]. Both *GLUA* and *PR1A* were expressed 10-fold more in trehalose-treated plants compared to water controls 6 h after treatment (Fig 5A) and their expression increased to over 100-fold at 48 h (Fig 5B). *PR1B* and *OSM*, which respond to the defense hormone ethylene (ET), were also significantly upregulated 48 h after trehalose treatment [62, 65]. The ET synthesis gene *ACO5* was also 10-fold more highly expressed compared to water controls 6 and 48 h after trehalose treatment (Fig 5A and 5B) [66]. Transcripts of the ABA-responsive genes *RD22* and *DHN_TAS* were slightly but significantly upregulated 48 h after trehalose treatment, possibly in response to the increased levels of ABA in xylem sap shown in Fig 5B [67–69]. Together, these qRT-PCR experiments revealed that tomato plants respond to trehalose by increasing expression of genes in JA, SA, ET, and ABA hormonal pathways involved in, among other things, biotic and abiotic responses.

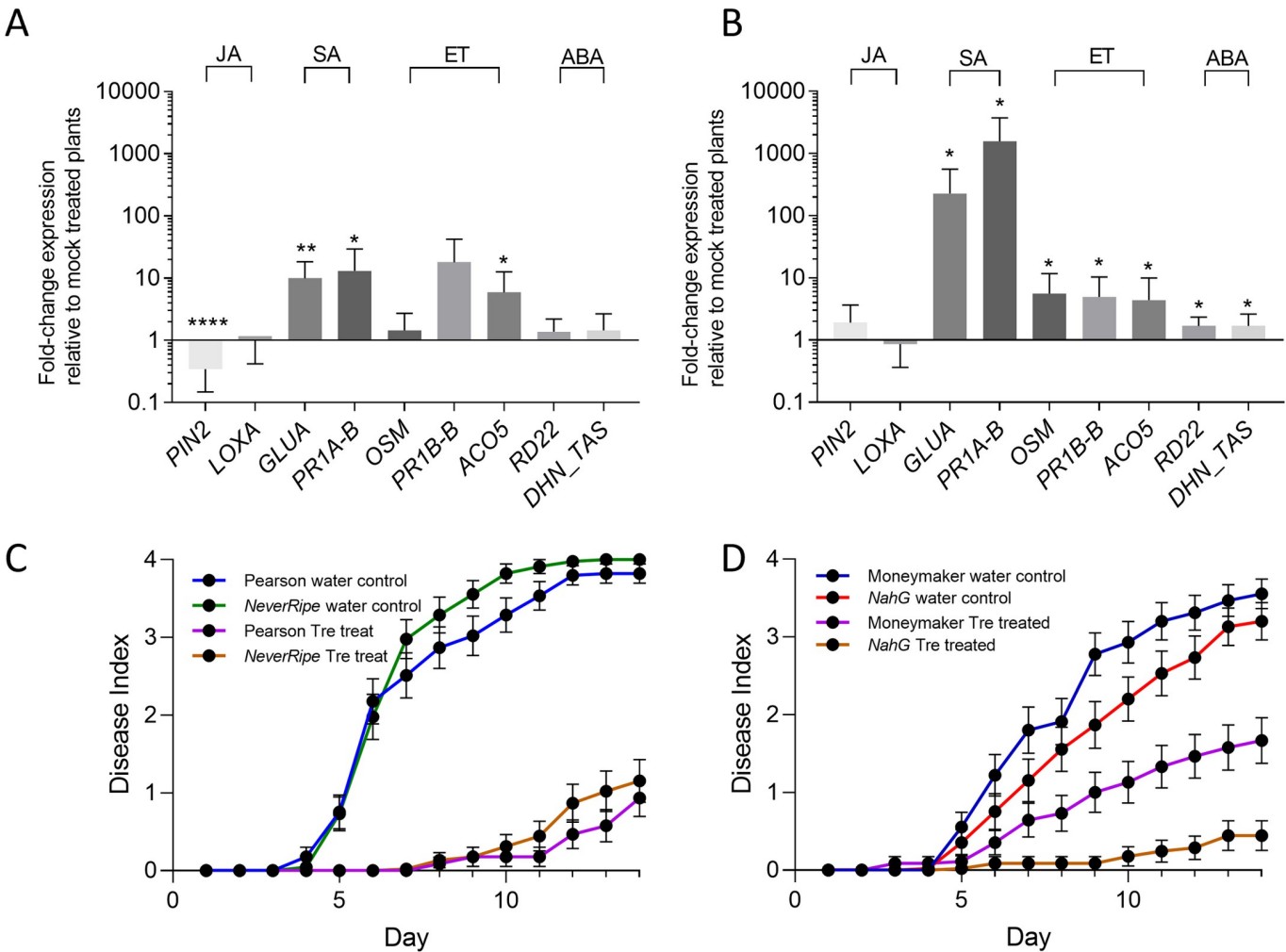

**Fig 5. Exogenous trehalose treatment affects expression of hormone-responsive tomato defense genes, and trehalose-mediated disease resistance is partially regulated by salicylic acid signaling.** Effects of trehalose treatment on expression of diverse defense genes in Bonny Best tomato stems, determined by qRT-PCR at **A)** 6 h and **B)** 48 h after trehalose treatment at the roots (one-sample t-test to a hypothetical mean of 1.0 where 1.0 = no difference in expression; asterisks: *, $P = .01-.05$; **, $P = .001-.01$; ***, $P = .0001-.001$; ****, $P < .0001$). Expression levels are shown as fold-change relative to those in mock (water)-treated plants. Data represent three biological replicates per time point, each containing five plants per treatment. Bars represent standard deviation. **C)** Wilt disease progress following *R. solanacearum* inoculation of water- or trehalose-treated *NeverRipe* ET-insensitive tomatoes and their parent cultivar 'Pearson' (one-way ANOVA of Area Under the Curve or AUC; 'Pearson' $H_2O$ vs. *NeverRipe* $H_2O$, $P = .91$; 'Pearson' tre vs. *NeverRipe* tre, $P = .98$). **D)** Wilt disease progress following *R. solanacearum* inoculation of water- or trehalose-treated SA-insensitive *NahG* over-expressing plants and their parent cv. Moneymaker (one-way ANOVA of AUC, Moneymaker $H_2O$ control vs. *NahG* $H_2O$ control, $P = .74$; Moneymaker tre vs. *NahG* tre, $P = .04$). For **C)** and **D)** plants were treated at the roots with trehalose, then inoculated with *R. solanacearum* via soil drenching 48 h later. Data represent three biological replicates, each containing 15 plants/treatment (45 plants total). Bars represent standard error.

The finding that SA- and ET-responsive tomato genes were upregulated 6 and 48 h after trehalose treatment suggested that ET perception and SA responses might be necessary for trehalose-mediated increase in bacterial wilt resistance. We tested this hypothesis by treating with trehalose the roots of ET-insensitive *NeverRipe* mutant tomatoes (cv. Pearson) and SA-degrading *NahG* transgenic tomatoes (cv. Moneymaker). After 48 h, all plants were soil drench inoculated with *R. solanacearum*. As observed for cv. Bonny Best, trehalose treatment increased bacterial wilt resistance of Pearson and Moneymaker plants. Trehalose also increased bacterial wilt resistance of both the *NeverRipe* and *NahG* plants compared to untreated controls (Fig 5C and 5D). However, trehalose-mediated bacterial wilt resistance does not require ET perception

since bacterial wilt developed similarly on trehalose-treated *NeverRipe* and the parental Pearson genotype ($P < .98$). In contrast, trehalose-treated *NahG* tomatoes were significantly more resistant to bacterial wilt than trehalose-treated Moneymaker parental line ($P < .04$). These results suggest that either a lack of downstream SA responses or overaccumulation of SA degradation products contributes to the trehalose-mediated increase in bacterial wilt resistance. Alternatively, trehalose may improve resistance downstream of SA responses.

More than half of the *NahG* tomato plants displayed a transient marginal leaf necrosis phenotype following trehalose treatment (data not shown), which could have been due to increased ROS production. To test this, we infused leaves from four-week-old wild type Bonny Best, Moneymaker, or *NahG* tomato plants with water, *R. solanacearum* cells, or 30 mM trehalose solution and observed ROS production with DAB staining. Infusing leaves with trehalose did not trigger excess ROS production (S4 Fig), suggesting that another physiological effect caused the transient marginal necrosis in the *NahG* plants.

## Trehalose treatment promoted accumulation of hormones and defense compounds in xylem sap

To better understand the mechanisms of trehalose-mediated wilt resistance, we measured levels of a suite of plant hormones, oxylipins, and defense-related compounds in xylem sap from infected and trehalose-treated Bonny Best tomatoes. We harvested sap from tomatoes displaying a disease index of 1 (DI = 1, partially wilted, late-stage infection) and from uninoculated plants at 6, 24, and 48 h after trehalose treatment. Using targeted quantitative mass spectrometry, we measured metabolites from the phenylpropanoid biosynthesis pathway, several oxylipins from the 9-lipoxygenase and 13-lipoxygenase pathways, and other selected defense metabolites (S2 Table).

Sap from *R. solanacearum*-infected plants and trehalose-treated plants contained different levels of targeted metabolites compared to sap from healthy tomatoes. Sap collected 24 h after trehalose treatment had the highest number of enriched compounds. These were consistent with a shock or wound response even though plants were not wounded [70]. Surprisingly, trehalose treatment, but not *R. solanacearum* infection, increased SA concentration at 24 h (Fig 6A). This result could mean that trehalose plays a role in SA signaling and suggested that SA might not have a significant role in xylem sap at late-stage disease. However, benzoic acid levels did increase in xylem sap from both trehalose treated and *R. solanacearum* infected plants (Fig 6B). Benzoic acid is a potent antimicrobial compound produced through the phenylpropanoid pathway that can be a direct precursor to SA [71, 72]. Interestingly, coumaric acid, another phenylpropanoid defense compound that *R. solanacearum* can degrade, was significantly increased in sap from *R. solanacearum*-infected plants but not in sap from trehalose treated plants (Fig 6C) [73]. Levels of traumatic acid, which results from the oxidation of traumatin produced by wounded plants, were higher in sap from both *R. solanacearum*-infected plants and trehalose-treated plants at 24 h (Fig 6D) [74].

The chemical signatures exclusively present in trehalose treated plants were the persistent increases in JA, the JA precursor 12-OPDA, and JA-Ile, the biologically active form of JA (Fig 6E–6G). This accumulation of jasmonates, together with the upregulation of JA and ET-pathway defense genes, suggested that trehalose treatment induces a systemic defense response that contributes to its protective effect [70, 75].

Curiously, several defense-related oxylipins from the reductase and lipoxygenase branches of the LOX pathway were enriched in sap from *R. solanacearum*-infected plants and 24 h post-trehalose treatment plants (Fig 6H–6K). These oxylipins include 9(*S*)-hydroxy-10(*E*),12(*Z*),15(*Z*)-octadecatrienoic acid (9-HOT, Fig 6H); 9(*S*)-hydroxy-10(*E*),12(*Z*)-octadecadienoic acid

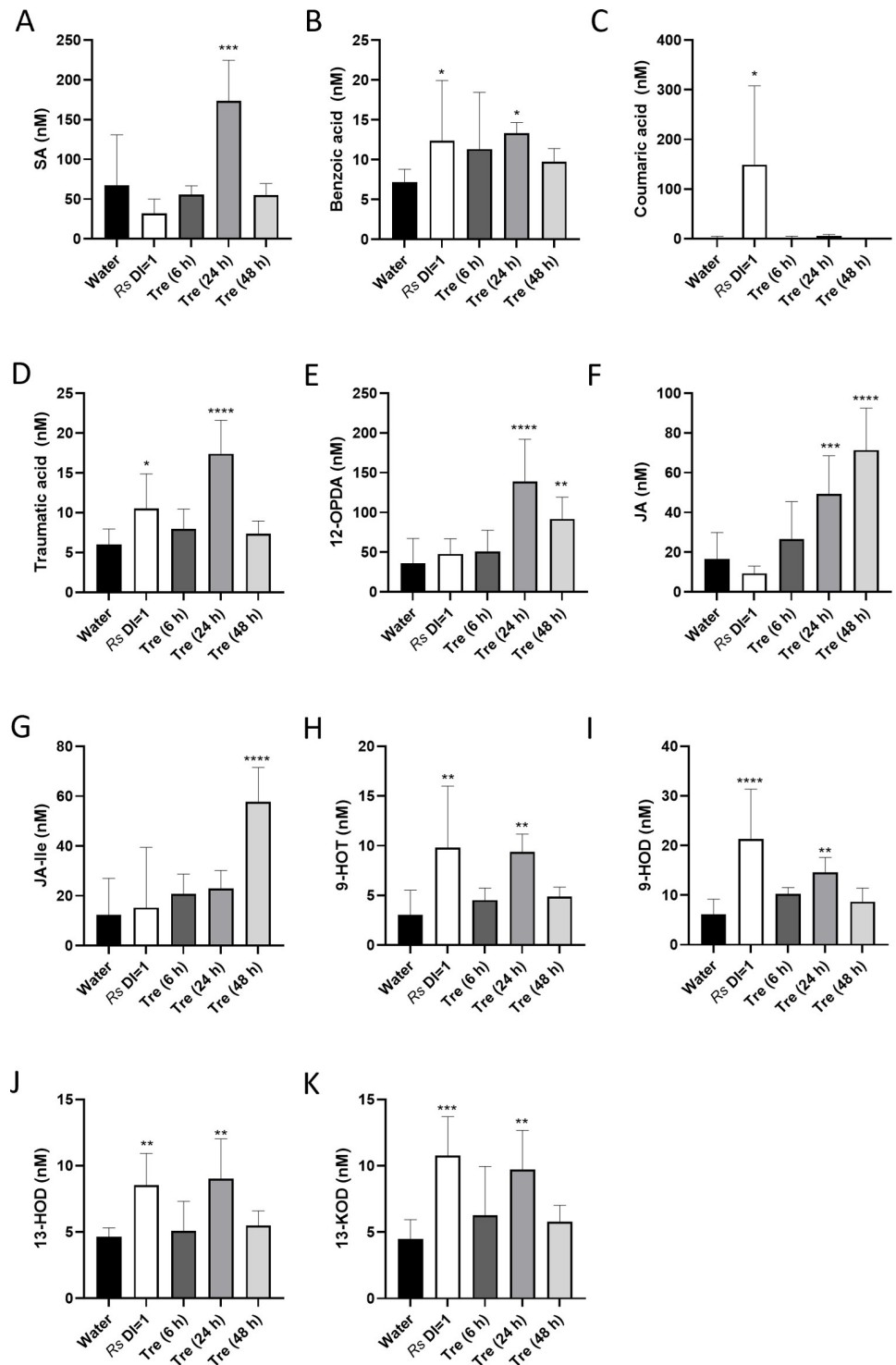

**Fig 6. Trehalose treatment increased metabolites related to defense and wounding in tomato xylem sap.** LC-MS/MS determined concentrations of hormones, oxylipins, and other metabolites in xylem sap collected from cv. Bonny Best tomato plants that were mock-inoculated ($H_2O$), inoculated with *R. solanacearum* GMI1000 when wilt symptoms first appeared (*Rs* DI = 1), or had been treated at the roots with 30mM trehalose (Tre (6h), etc.) Differences from water-treated control plants were determined by one-way ANOVA Fisher's LSD multiple comparison test: *, $P = .01$-$.05$; **, $P = .001$-$.01$; ***, $P = .0001$-$.001$; ****, $P < .0001$. Data shown represent the means of samples from 4–6 individual plants. Bars indicate mean, vertical lines show standard deviation. **A)** SA = salicylic acid; **B)** BA = benzoic acid; **C)** COUMA = coumaric acid; **D)** TA = Traumatic acid; **E)** 12-OPDA = 12-oxo-8(*Z*),13

(*Z*)-phytodienoic acid; **F)** JA = jasmonic acid; **G)** JA-Ile = jasmonic acid isoleucine; **H)** 9-HOT = 9(*S*)-hydroxy-10 (*E*),12(*Z*),15(*Z*)-octadecatrienoic acid; **I)** 9-HOD = 9(*S*)-hydroxy-10(*E*),12(*Z*)-octadecadienoic acid; **J)** 13-HOD = 9 (*S*)-hydroxy-10(*E*),12(*Z*)-octadecadienoic acid; and **K)** 13-KOD = 13-oxo-9(*Z*),11(*E*)-octadecadienoic acid.

(9-HOD, Fig 6I); 13(S)-hydroxy-9(Z),11(E)-octadecatrienoic acid (13-HOD, Fig 6J); and 13-oxo-9(*Z*),11(*E*)-octadecadienoic acid (13-KOD, Fig 6K). All these oxylipins have antimicrobial and signaling activity in defense against pathogens [76–78]. Induction of these oxylipins in this study suggests involvement in resistance to bacterial wilt. Altogether, trehalose treatment had an unexpectedly large impact on xylem levels of defense metabolites, particularly jasmonates and SA, which have often been considered antagonistic signals.

## Discussion

To survive drought, plants must respond to and overcome multiple adverse physiological events [68, 79]. Our stringent meta-analysis of four different transcriptomic studies found that plants infected with *R. solanacearum* have a gene expression signature like that of water-stressed plants, suggesting that bacterial wilt disease physiologically mimics abiotic drought stress (Fig 1). The consistency of this response across multiple tomato cultivars suggest that this is a robust, replicable phenomenon.

Infected tomatoes slightly upregulated TPS7 in our system and others (Fig 1) [51], suggesting this gene could be responsible for the observed increased trehalose synthesis during bacterial wilt. This gene was also upregulated 36 h after infection with *P. syringae and* silencing this gene (annotated as *SlTPS3* in that study) decreased tomato trehalose content in response to the fungal pathogen *Botrytis cinerea* and also decreased plant resistance to *B. cinerea* [51].

In response to *R. solanacearum* infection, tomatoes downregulated multiple aquaporins (Fig 1), suggesting that plants may try to alter water flow during bacterial wilt. In particular, SlTIP2;2 (Solyc03g120470), an aquaporin that improved transpiration and yield when overexpressed in tomato plants, was notably downregulated in our meta-transcriptomic analysis [80]. Accordingly, *Arabidopsis* PIP family aquaporin proteins were generally downregulated during drought stress [81]. Tomato aquaporins are encoded by diverse families and channel other solutes in addition to water, so it would not be expected that all aquaporins play the same role during bacterial wilt [82, 83].

An ABA-inducible dehydrin and an ABA hydroxylase gene were strongly upregulated in infected plants (Fig 1). In *Arabidopsis* and tomato, CYP707A ABA hydroxylases, which catalyze the first step in ABA degradation, are involved in dehydration and rehydration [84–87]. This suggests both that tomatoes tightly regulate ABA catabolism during bacterial wilt and that bacterial wilt causes water stress. Expression of TAS14, a dehydrin inducible by environmental stress and ABA, was highly upregulated in infected plants, further evidence that bacterial wilt causes drought stress [88]. *Arabidopsis* ABA synthesis, signaling, and responsive genes were similarly affected during late-stage bacterial wilt infection [89]. Elevated ABA levels are usually associated with drought stress; together with the observation that *R. solanacearum* infection lowers plant stomatal conductance, this indicates that bacterial wilt disease significantly disrupts tomato water physiology. Further characterization of ABA-responsive networks in tomatoes could identify targets for breeding more hydraulically efficient tomatoes that are also more resistant to bacterial wilt.

Our data suggest that there might be an indirect defensive component to ABA concentration and signaling. The levels of ABA in xylem sap increased in response to both *R. solanacearum* infection and trehalose treatment (Fig 2A), and our transcriptomic data suggest tomato plants degrade ABA during disease via an ABA hydroxylase (Fig 1). Although it would

be interesting to test the effects of trehalose and *R. solanacearum* on tomato ABA synthesis mutants, existing ABA mutants (*flacca*, *notabilis*, *sitiens)* share a wilting phenotype due to ethylene overproduction and lack of control over stomatal closure. This physiological wilting interferes with assessing pathogen-caused wilting symptoms [90–92]. However, *Arabidopsis* studies also indicate that ABA is important for tomato resistance to wilt disease. Deleting several cellulose synthase (*irx*) genes increased *Arabidopsis* resistance to bacterial wilt in an ABA-dependent manner and ABA synthesis mutants were more susceptible to *R. solanacearum* [93]. Further, ABA perception mutants are also hyper-susceptible to bacterial wilt [94].

Treating tomato roots with ABA reduced both stomatal conductance (Fig 2B) and *R. solanacearum* colonization of stems (Fig 2C), suggesting that *R. solanacearum* spread depends, at least in part, on active hydraulic conductance and a negative pressure gradient in host xylem. It has long been assumed that *R. solanacearum* uses the plant's own water transport to spread within its host, but this is now demonstrated experimentally. When this pathogen blocks host plant water flow, it decreases its own spread and pathogenic fitness. Interestingly, some wilt-resistant tomato varieties thrive even when they are latently infected. This supports the theory that resistant hosts can, to some extent, locally limit vascular bacterial growth enough to prevent occlusion of xylem vessels [95]. We speculate that ancestors of *R. solanacearum* were once endophytes, but lost the ability to moderate their population growth in susceptible hosts, possibly as a result of evolutionary selection pressures imposed by agriculture, as has been proposed for *Xylella fastidiosa* [96].

In our system, exogenous trehalose temporarily improved water conservation in uninfected tomatoes and delayed abiotic wilting of drought-stressed plants (Figs 3A and 4A). This was likely because trehalose treatment temporarily reduced stomatal conductance and transpiration (Fig 3C and 3D) and increased water use efficiency (S2A Fig). Trehalose increased ABA content in xylem sap 24 h post-treatment (Fig 3B), and disease symptoms often occurred after ABA concentration peaked, consistent with a potential signaling role for ABA. These results confirm and extend findings that trehalose, ABA signaling, and water use interact in complex ways. Additional studies are needed to connect ABA signaling and concentration in xylem sap to downstream effects on disease resistance.

Other *Arabidopsis* and tomato trehalose/water stress studies have confirmed some of the phenotypes we observed. The *Arabidopsis TPS1* gene, which is responsible for maintaining T6P levels, responds to ABA and helps regulate stomatal aperture [97, 98]. At least two *Arabidopsis* TPP proteins (AtTPPI and AtTPPE] that are controlled by ABA-responsive transcription factors contribute to trehalose production and mitigate drought stress [99–101]. Yu and co-workers found that applying trehalose to tomato roots improved water conservation, decreased ABA biosynthesis gene expression, increased expression of ABA-responsive genes, kept stomates closed five days after trehalose treatment without water, and increased ABA levels in leaves after 24 h [42]. Our results contrast in some ways with this paper. We found that ABA content temporarily increased in xylem sap 24 h after trehalose treatment (Fig 3B), while stomatal conductance was reduced for less than 24 h (Fig 3D). These differences may be explained by methodological variation: Yu and co-workers did not include non-drought stressed tomato controls, they measured ABA in leaves rather than xylem, and they measured stomatal aperture rather than conductance. We observed a lag between the stomatal conductance response in leaves and ABA concentration in stem xylem sap, leading us to hypothesize that ABA concentration in leaves would be a more biologically relevant marker for behavior of leaf stomates. It would be interesting to determine if either exogenous trehalose or ABA treatment changes trehalose concentration in tomato xylem sap.

In our system, trehalose protected plants against disease when applied via soil drenching, but not when applied as a spray (Figs 4A and S3B). Spraying wheat leaves with trehalose did

increase plant growth, improve water stress tolerance, and provide resistance to powdery mildew, a leaf disease [102–104]. Multiple applications of trehalose to leaves via spray also increased tobacco growth under nitrogen limiting conditions [105]. We found no studies demonstrating that spraying tomato leaves with trehalose promotes resistance to pathogens, although leaf infiltration provides resistance to aphid predation [106]. These findings suggest that all plants do not respond to exogenous trehalose in the leaves in the same way. Additionally, the receptors for sugar-mediated DAMP response in tomatoes are unknown; grafting experiments may offer insights into whether these are primarily expressed in the roots.

The plant hormone SA, which mediates resistance to *R. solanacearum* in many systems, is also necessary for induction of plant innate immunity and activation of systemic acquired resistance [66, 107–111]. We found SA-responsive tomato defense genes were highly upregulated 6 and 48 h after trehalose treatment (Fig 5A and 5B). These same genes are upregulated in diseased tomato plants following *R. solanacearum* soil soak infection, suggesting trehalose treatment activates SA-mediated resistance pathways that could affect *R. solanacearum* infection [112]. Expression of the SA-responsive marker gene PR1b is affected in several tomato trehalose metabolism knockdown mutants (decreased in *SlTPS4*, increased in *SLTPS5* and *SlTPP2*) after *P. syringae* pv. *tomato* challenge [51], further suggesting interplay between trehalose metabolism and resistance to bacterial pathogens.

Trehalose treatment increased wilt resistance of SA-degrading *NahG* tomato plants, but not of the parental Moneymaker cultivar (Fig 5D). There are several possible explanations for this result. First, trehalose treatment could have increased SA levels so much that overproduction of catechol, the breakdown product of SA, contributed to resistance [113]. Second, the excess ET and JA in *NahG* plants could promote resistance to *R. solanacearum*. After infection with *P. syringae* DC3000 *NahG* tomatoes produce more ET and JA, both of which can contribute to disease resistance [114, 115].

ET is involved in *R. solanacearum*-host interactions and ET-responsive defense genes were upregulated at 6 and 48 h post trehalose treatment (Fig 5A and 5B). Interestingly, ET and JA signaling-responsive defense-related genes were downregulated in tomatoes with silenced *SlTPS3*, *4*, and *7* [51]. However, ET-insensitive *NeverRipe* tomatoes were neither more nor less resistant to *R. solanacearum* than wild type after trehalose treatment (Fig 5C). Treatment with a biocontrol strain of the oomycete *Pythium* enhanced resistance to *R. solanacearum* in *NahG* tomatoes, but not in a JA mutant [116]. Since the ET defense pathway appeared less relevant in our study, it may be useful to determine if JA enhances *R. solanacearum* resistance in tomatoes. Our experiments with tomato defense hormone mutants also demonstrated that trehalose treatment significantly delays disease incidence relative to water-treated control plants in multiple tomato genotypes.

Plants use hormones to fine-tune their growth, development, and defense, and xylem vessels can act as signal highways for long-distance transport of auxin, ABA, JA, and SA [117, 118]. We observed increased SA concentration in xylem sap of trehalose-treated plants but not in sap from *R. solanacearum*-infected plants (Fig 6A). This was surprising because SA signaling and concentration are integral to plant response to bacterial wilt [66]. However, *R. solanacearum* actively degrades SA, which is toxic to the pathogen [119], so SA concentrations might be higher in root xylem than in stem xylem. For instance, an *Arabidopsis* mutant impaired in secondary wall deposition had higher SA concentrations in the root and increased resistance to *R. solanacearum* infection, possibly through vascular immunity [120]. Taken together, the increased SA levels in xylem sap (Fig 6A), the increased expression of SA-dependent defense genes (Fig 5A and 5B), and the trehalose-dependent *NahG* resistance phenotype (Fig 5D) suggest that the protective effect of trehalose requires systemic resistance induced via SA.

Unexpectedly, jasmonate levels were elevated in trehalose-treated plants but not in *R. sola-nacearum*-infected plants (Fig 6E–6G). As discussed earlier, there is some evidence that JA can play a role in *R. solanacearum* defense. Genes related to ET, JA, and auxin signaling were upregulated in resistant LS-89 tomato seedlings one day after stem inoculation with *R. solanacearum* [121]. The higher concentration of jasmonates in trehalose-treated xylem sap may have contributed to resistance. This hypothesis could be tested using JA-deficient tomato mutants.

SA- and JA-mediated plant defense responses are often antagonistic [122, 123]. Our gene expression analysis followed this pattern; at 6 h post trehalose treatment JA-responsive *pin2* was downregulated, while SA-responsive *PR1b* and *gluA* were upregulated (Fig 5A). Roots of wilt resistant Hawaii7996 tomatoes displayed a similar gene expression profile 24 h after *R. solanacearum* infection [57]. Despite the antagonism suggested by the gene expression data, the concentrations of SA, JA, and the JA precursor 12-OPDA all increased in trehalose-treated plants at 24 h (Fig 6A and 6E–6G). In maize, levels of OPDA and JA are irrelevant for the classic SA/JA-Ile antagonism and 12-OPDA and SA are co-regulated [124]. Infusing maize xylem sap with 12-OPDA increased resistance to the fungal pathogen *Colletotrichum graminicola* while infusing similar concentrations of JA increased susceptibility. Disease experiments with JA-insensitive tomato mutants could determine if 12-OPDA or JA contribute to resistance and define their relationship to SA signaling.

JA signaling also induces leaf senescence, and chlorosis is associated with increased JA production [125]. Consistent with this, trehalose treatment significantly reduced chlorophyll content 2–10 days post-treatment (S2E Fig), which correlated with the increased JA production in trehalose-treated plants. Similarly, applying 5 mM trehalose to soybean roots did not affect JA concentration, but it did decrease chlorophyll content at ten days [126].

Trehalose treatment and *R. solanacearum* infection both increased xylem sap levels of several antimicrobial compounds, notably phenolics (Fig 6B and 6C). *R. solanacearum* requires at least two multi-drug efflux pumps and a hydroxycinnamic acid degradation pathway for full virulence and fitness, showing that antimicrobials, including phenolics, protect tomato plants from bacterial wilt [73, 127]. Plants have specialized cells that store and release phenolics into the xylem as part of their defense response [128], and our data indicate that trehalose treatment and *R. solanacearum* infection both induce this release.

Levels of the oxylipins 9-HOT, 9-HOD, 13-HOD, and 13-KOD increased in sap from both *R. solanacearum*-infected plants and trehalose-treated plants 24 h after treatment (Fig 6H–6K). These metabolites, which accumulate in wounded maize, bean, and *Arabidopsis*, can be potent inducers of defense gene response [129–131]. Further, these oxylipins have direct antimicrobial activity against fungal and bacterial pathogens [76]. The data generated by this study deserve further analysis since little is known about oxylipin signatures during bacterial wilt or if oxylipins are toxic to *R. solanacearum*. However, 9-LOX-derived oxylipins, including 9-HOT and 9-HOD, increase in tobacco leaves inoculated with *R. solanacearum* GMI100, which triggers an intense defensive hypersensitive response (HR) on tobacco [132]. We speculate that the presence of related oxylipins in our study may indicate vascular HR. It would be interesting to see if defense compound concentration changed in the xylem sap of resistant tomatoes challenged with *R. solanacearum*.

We propose that tomato plants activated induced systemic resistance because they recognized trehalose as an endogenous (self-derived) damage associated molecular pattern or DAMP [133]. DAMPs are products released during pathogen invasion or wounding that alert the plant to an attack [134–136]. A DAMP can be almost anything that is usually intracellular like DNA, sugars, or NAD (indicating cellular leakage), or products of lignin and cellulose degradation (indicating mechanical or enzymatic disruption of plant cell walls). Plant systemic

resistance response to DAMPs, mechanical wounding, or pathogen challenge typically involves accumulation of SA, JA, and defense oxylipins, and increased expression of SA, JA, and ET-responsive defense genes [70, 75, 124, 137]. Consistent with this definition, we found that treating tomato roots with trehalose increased SA, JA, and defense oxylipins (Fig 6), and upregulated SA, JA, and ET defense genes (Fig 5A and 5B). Our finding adds to the growing evidence that sugars, specifically trehalose, contribute to plant innate immunity [138].

Could treating roots with compatible solutes like trehalose in the field provide temporary resilience until disease or drought pressure have passed? Other chemical inducers of plant defense responses can reduce bacterial wilt. Acibenzolar-S-methyl (Actigard) and DL-3-aminobutyric acid (BABA) both lowered wilt disease incidence in greenhouse and field by promoting defense gene expression and defense compound production, respectively [139–141]. Additionally, growing tomatoes hydroponically with SA and JA increased defense enzyme production and reduced bacterial wilt incidence [142]. Finally, exogenous L-histidine induced resistance to bacterial wilt in an ET-dependent manner in *Arabidopsis* and tomato [143]. Further studies are needed to know if treating tomatoes with exogenous trehalose could be a practical strategy for bacterial wilt management.

*R. solanacearum* GMI1000 produces and secretes abundant amounts of a trehalase (TreA) during tomato infection. Trehalose is not an important nutrient for the pathogen but the ability to degrade trehalose contributes to bacterial fitness and virulence by an unknown mechanism [29, 144]. Interestingly, treating plants with the trehalase inhibitor Validamycin A increased bacterial wilt resistance [48]. We previously showed that the trehalose in xylem sap of *R. solanacearum*-infected tomato plants is produced by the plant [28]. Could *R. solanacearum* secrete trehalase to counter the effects of plant-produced trehalose?

Breaking down plant-produced trehalose could benefit *R. solanacearum* in several ways. First, the pathogen may degrade xylem sap trehalose in order to reduce host plant defenses by removing this potential DAMP. Second, as discussed above, trehalose reduces tomato transpiration, and transpiration helps *R. solanacearum* spread and colonize host xylem. *R. solanacearum* may degrade trehalose to maintain this disseminating sap flow by preventing trehalose-mediated stomatal closing. Third, *R. solanacearum* could degrade trehalose to interfere with the trehalose-T6P signaling that plants use to fine-tune their sucrose to starch ratios. Trehalose degradation could thus indirectly increase the amount of available sucrose, which is a key carbon source fueling *R. solanacearum* growth in xylem [29, 144]. The fact that the *R. solanacearum* effector RipTPS is a functional trehalose-6-phosphate synthase is consistent with the hypothesis that this pathogen benefits from manipulating host plant trehalose metabolism [145]. Additional experiments are needed to test these non-exclusive hypotheses about how trehalase benefits *R. solanacearum*.

## Conclusion

This study argues that trehalose is present in xylem sap of *R. solanacearum*-infected tomatoes as a response to wilt disease-mediated water stress. Both trehalose levels and the expression of water stress genes changed in response to *R. solanacearum* infection. Further, ABA content, an indicator of drought status, increased in xylem sap of diseased plants. We confirmed the link between trehalose and ABA signaling and drought stress in plants by demonstrating that trehalose treated plants temporarily reduce stomatal conductance, contain more ABA in their xylem sap, and are more drought tolerant.

When applied to plant roots, trehalose reduced bacterial wilt incidence and reduced host xylem conductance, which aids bacterial colonization. SA-responsive defense genes were upregulated in trehalose-treated plants, SA non-responsive *NahG* plants were more resistant to *R.*

*solanacearum* after trehalose treatment, and SA concentration transiently spiked in xylem sap from treated plants. JA, other oxylipins, and many antimicrobial compounds also increased in xylem sap of trehalose-treated plants. Taken together, these results suggest that plants perceive trehalose as a DAMP and respond with systemic disease resistance. In turn, *R. solanacearum* degrades trehalose as a counter-defense.

## Methods

### Total RNA extraction and RNA-seq sample preparation

RNA extraction and RNA-seq sample preparation for the transcriptomic studies were performed as described [57, 146].

### RNA-seq data analysis

Illumina HiSeq2500 paired-end 100-bp; Illumina HiSeq2500 paired-end 150 bp and Illumina HiSeq2000 paired-end 100-bp RNA sequencing were performed on the 18, 6, and 12 samples from three independent studies (two previously unpublished [Studies A+B], and [Study C] French, Kim et al. 2018). RNA-sequencing data were analyzed through the same analysis pipeline. Read quality was controlled using FastQC version 0.11.9 [147]. Reads were trimmed to remove low-quality sequences and the Illumina sequencing adaptors (TruSeq3-PE-2) using Trimmomatic version 0.39 [148]. After trimming and mapping, a total of 930,004,276, 135,532,392, and 103,831,164 paired-end reads were generated for Studies A, B, and C, respectively. Reads for each sample were mapped to the ITAG4.0 *S. lycopersicum* reference genome using STAR version 2.7.4.a [149]. Library type was set to strand-specific (reverse) for Study A and Study B, and set to no-stranded in Study C. Gene expression was measured as the total number of reads for each sample uniquely mapped to the reference using HTSeq version 0.11.1 and samtools version 1.10 [150, 151]. Each sample averaged about 12, 9, and 4 million uniquely aligned reads for Studies A, B, and C, respectively. "EdgeR" package version 3.28.0 was used to filter for low counts such that at least three of the samples in each study had at least three counts per million, resulting in a total of 19123, 18193, and 18917 genes remaining for Studies A, B, and C, respectively [152].

Differential gene expression within each study was conducted using the DESeq2 package Bioconductor version 3.11 and a Benjamini-Hochberg false discovery rate multiple testing correction of $P = .05$ [153, 154]. Meta-transcriptomic analysis was performed using the metaRNA-Seq package version 1.0, and differential gene expression within the meta-analysis was determined with the DESeq2 results, Fisher, and inverse normal techniques with a $P = .05$, as described [155]. Differentially expressed genes were filtered using a log2 fold change $>|0.585|$ (corresponds to a fold change of $|1.5|$).

Gene Ontology analysis was performed using the ShinyGO analysis tool version 0.61, agriGO version 2.0, and PANTHER GO analysis tool version 14 [156–158]. *Arabidopsis thaliana* homologs of S. *lycopersicum* genes coding for transcription factors were identified using Phytozome platform version 12.1 [159]. Heatmaps were visualized with R software version 3.6.1 package "pheatmap" version 1.0.12 [160, 161]. Boxplots were visualized using package "ggplot2" [162].

Candidate TPS and TPP proteins were identified in the most recent annotations of the tomato (cv. 'Heinz 1706') genome available on NCBI and the Sol Genomics Network (ITAG4.0, https://solgenomics.net/tools/blast/, accessed 03/2020) using BLASTP with standard settings [163]. BLASTP protein sequences were derived from the *E. coli* K-12 *otsA* and *otsB* proteins, as well as *Arabidopsis* trehalose-related proteins [35, 36, 52]. Tomato cv. Heinz 1706 genome contained ten TPS genes, eight TPP genes, and one trehalase gene. These genes were previously

identified as summarized [51]. Tomato ABA-related, LEA, and aquaporin genes were identified by their PANTHER annotation, and many were previously annotated [82, 164].

The transcriptomic studies were performed and labeled as follows: S1, roots of wilt-susceptible West Virginia 700 tomato plants (WV) sampled 24 h after infection with *R. solanacearum* strain K60; S2, roots of WV sampled 48 h after infection with *R. solanacearum* strain K60; S3, seedling roots of wilt-susceptible cv. Bonny Best (BB) sampled 24 h after infection with *R. solanacearum* strain GMI1000; S4, mid-stems of BB sampled 72 h after inoculation with *R. solanacearum* strain GMI1000.

## Hormone analysis leaf and sap collection

For the hormone analysis, xylem sap was harvested and collected as described [16]. Unwounded three-week-old wilt-susceptible Bonny Best plants were inoculated with wild-type *R. solanacearum* or water via naturalistic soil soak in which 50 mL of $1x10^8$ CFU/mL were poured into a pot containing one plant. Plants were harvested when they reached DI = 1 (1–25% of leaf area wilted). Trehalose-treated plants were treated with 50 mL of 30 mM trehalose solution or water, then harvested 6, 24, or 48 h later. Xylem sap was harvested by de-topping plants 1 cm above the cotyledon and collecting the exuded sap over 30 minutes. The sap that seeped within the first five minutes was discarded to reduce phloem contamination. Sap was kept on ice. Bacteria were removed from sap via chilled centrifugation, and sap was subsequently flash frozen. Stems were destructively sampled to quantify *R. solanacearum* colonization as CFU/g stem by serial dilution plating. Xylem sap was normalized by sap volume (250–500 μL) and concentration of bacteria in the stem. Each test group contained six technical replicates (individual plants).

For metabolite analysis of sap, 95 μL of xylem sap was mixed with 5 μL of 50 μM of internal standard consisting of; d-ABA ($[^2H_6](+)$-*cis*, *trans*-abscisic acid, Olchem, Olomouc, Czech Republic), d-CA ($d_7$-CA, *trans*-cinnamic acid), d-IAA ($[^2H_5]$ indole-3-acetic acid, Olchem), d-JA (2,4,4-$d_3$; acetyl-2,2-$d_2$ jasmonic acid, CDN Isotopes, Pointe Claire, Quebec, Canada), and d-SA ($d_6$-SA; Sigma-Aldrich, St Louis, MO, USA). Samples were transferred into autosampler vials for LC-MS/MS analysis. Authentic analytical external standards for each metabolite were used to identify select ions and retention times for each metabolite (S3 Table). The simultaneous detection of several phytohormones used the following methods with modifications [165]. We used an Ascentis Express C-18 Column (3 cm × 2.1 mm, 2.7 μm) (Sigma-Aldrich, St. Louis, MO, USA) connected to an API 3200 LC-MS/MS (Sciex, Framingham, MA, USA) using electrospray ionization in negative mode with multiple reaction mentoring. The injection volume was 10 μL and had a 500 μL $min^{-1}$ mobile phase consisting of Solution A (0.2% acetic acid in water) and Solution B (0.2% acetic acid in acetonitrile) with a gradient consisting of (time–%B): 0.5–10%, 1.0–20%, 21.0–70%, 24.6–100%, 24.8–10%, 29 –stop. After integration with Analyst v1.6.3 (Sciex), the concentration of endogenous metabolites was determined by comparing their peak areas to those of isotopically labeled internal standards using calculated response factors.

## Stomatal conductance, transpiration, and photosynthesis measurements with LI-COR

To assess the effect of trehalose treatment on tomato stomatal conductance ($g_s$), transpiration (E), and photosynthesis ($A_{net}$), we used a LI-COR portable photosynthesis system (model LI-6400XT, LI-COR Biosciences, Lincoln, NE, USA) equipped with a $CO_2$-controller and a red-blue LED chamber (model LI-6400-02b). The ambient $CO_2$ concentration was set to 400 ppm, flow rate to 300 μmol $s^{-1}$, and the chamber light source set at an intensity of 1200 μmol $m^{-2} s^{-1}$

for all measurements. Fifteen-day-old Bonny Best plants were treated with 50 mL of trehalose solution or water, then watered every day after for the duration of the experiment. LI-COR measurements were taken at 10min, 6 h, 24 h, 48 h, and 72 h post trehalose treatment. The experiment was repeated twice with ten plants per treatment.

### *R. solanacearum* culture conditions

*R. solanacearum* strains were cultured from water stocks or -80°C glycerol stocks on TZC plates containing casamino acids, peptone, glucose, and tetrazolium chloride, and incubated at 28°C for 48 h [166]. *Rs ΔtreA* cultures were supplemented with 20 mg L$^{-1}$ spectinomycin. The kanamycin-resistant variant of wild-type GMI1000 was supplemented with 25 mg L$^{-1}$ kanamycin on plates and in liquid culture. Overnight cultures of *R. solanacearum* were grown at 28°C in CPG in a shaking incubator.

### ABA treatments

We assessed the effect of ABA treatment on two-week-old tomato stomatal conductance and subsequent bacterial wilt incidence. ABA was prepared at a stock concentration of 50mg/mL in DMSO. The water + DMSO control plants were watered daily with 50 mL of water supplemented with the volume of DMSO used to make the 100 μM ABA solution. The ABA-treated plants were watered daily with 50 mL of 50 or 100 μM ABA solution in water. After 48 h of ABA treatment, plants were soil drenched with 50 mL of a 1x10$^8$ CFU/mL of *R. solanacearum* GMI1000 with kanamycin resistance (to aid in recovery from the plant) [73]. Bacteria were resuspended at the desired O.D. in water + DMSO or the ABA solutions, so plants were inoculated and ABA treated concurrently. Tomatoes were destructively sampled five days post-infection (after 168 h of ABA treatment) to assess bacterial colonization of stems. Stomatal conductance, transpiration, and photosynthesis measurements were taken for the entirety of the experiment with the LI-COR as described 4–6 h after watering/ABA treating in the morning, which was the length of time it took ABA treatment to affect conductance.

We assessed ABA toxicity to *R. solanacearum* by supplementing CPG broth with 50–100 μM ABA or the equivalent volume of DMSO, and growing *R. solanacearum* cells in this media for 48 h. Cells were grown in a shaking 96 well plate at 28°C in a BioTek Synergy HT plate reader, which measured Abs$_{600}$ every half hour. The experiment was performed once with eight technical replicates per treatment.

### Plant assays with *R. solanacearum*

Plant growth conditions, colonization, and disease curve assays were performed as described [28, 167]. All plant assays were grown on 12 h light/dark cycles at 28°C with ~60% relative humidity unless stated otherwise. For the disease progress assays with trehalose treatments, fifteen-day old Bonny Best tomatoes were treated with either water (control) or 50 mL of 30 mM trehalose solution. Plants were not watered 24 h post-treatment per standard procedure for soil soak inoculation assays. At 48 h post-treatment plants were soil drenched with 50 mL of 1x10$^8$ CFU/mL of wild type *R. solanacearum* GMI1000 (or *ΔtreA* for the trehalose treatment experiments with this mutant). For the petiole inoculations, 3-week-old tomatoes were treated with trehalose or water as described above; after 48 h a true leaf was removed, and 2000 wild type GMI1000 cells in 10 μL of water were pipetted onto the cut stem. To assess the effect of spraying trehalose on bacterial wilt incidence, fifteen-day-old Bonny Best plants were either treated with 30 mM trehalose solution through the soil as described or leaves were sprayed with trehalose solution (or water) to run-off. For all assays, plants were inoculated with *R. solanacearum* 48 h post trehalose root or leaf treatment.

After treatments and inoculations, all plants in disease assays were watered on alternating days post-inoculation with half-strength Hoagland's solution. Disease assay plants were rated daily on a 0–4 disease index scale where 0 = no symptoms, 1 = 1–25% of leaf area wilted, and 4 = 76–100% of leaf area wilted or dead. All disease curves represent four biological replicates, each containing 15 plants/treatment.

To assess *R. solanacearum* colonization of trehalose treated plants, fifteen-day-old Bonny Best plants were treated with 50 mL 30 mM trehalose solution or water. 48 h post-treatment, plants were inoculated with 50 mL of $1x10^8$ CFU/mL wild type *R. solanacearum* GMI100. At 5, 6, and 7 dpi plant stems were destructively sampled 1 cm above the cotyledon, ground, and serially dilution plated.

*NahG* salicylic insensitive tomato seeds ('Moneymaker' background) and *NeverRipe* ET-insensitive seeds ('Pearson' background) were propagated in our lab [168, 169]. Disease assays with trehalose-treated hormone insensitive plants were as described above.

## Water conservation

Two-week-old Bonny Best plants grown as described above were treated with trehalose or water. Water was subsequently withheld and plants were assessed for wilting symptoms, using the same visual index scale as for disease progress assays. The experiment included four biological replicates, each with ten plants per treatment.

## QRT-PCR measurement of tomato trehalose and defense gene expression

JA, SA, and ET defense gene primers were previously characterized in tomato *R. solanacearum* response (S1 Table) [112]. The *rd22* and *dhn_tas* ABA-responsive primers were previously described [67, 69]. *ACTIN* and *DNAJ* primers served as our constitutively expressed normalizing genes.

To measure defense gene expression, two-week-old Bonny Best tomatoes were treated with 50 mL of 30 mM trehalose or water. 100 mg of stem tissue was harvested 6 or 48 h post-treatment, then flash frozen. For all samples, stems were homogenized for a minute in a Powerlyzer (Qiagen) at room temperature in 450 µL of Buffer RLT from the RNAeasy Plant Mini Kit (Qiagen). RNA extraction proceeded as specified in the kit protocol with off-column DNase I treatment using a DNA-free kit (Thermo-Fisher). RNA quality and quantity were assessed with a NanoDrop (Thermo-Fisher) and reverse transcribed into cDNA using Superscript III First-Strand Synthesis SuperMix (Invitrogen). Lack of genomic DNA contamination was confirmed with PCR of no reverse transcriptase controls. Quantitative real-time PCR was performed in 10 µL reactions with EvaGreen Mastermix (Biotium) using kit-specified reaction parameters in a Quant Studio 5 Real-Time PCR machine (ThermoFisher Scientific). Averaged actin and *dnaJ* values were used as the stably expressed normalization genes. Fold change values were calculated using the $\Delta\Delta C_T$ method as described [112]. The data are presented as fold change in gene expression compared to mock-inoculated controls and represent 16 plants per treatment over four biological replicates.

## Relative water content

To assess the water status and osmotic adjustment of tomato plants treated with trehalose, we measured relative water content (RWC) in leaves. This directly measures cellular hydration. Three-week-old Bonny Best tomato plants were treated with 50 mL of water or 30 mM trehalose solution. 48 h post-treatment, leaves were excised at the petiole and weighed for fresh weight. Leaves were then rehydrated by floating on distilled water in closed petri dishes for 4 h, at which time they were blotted, and their turgid weight was assessed. Leaves were then

thoroughly dried in a food dehydrator oven for 24 h, and dry weight was evaluated. RWC was calculated as follows: % RWC = [(Leaf fresh weight–dry leaf weight)/(turgid leaf weight-leaf dry weight)] x 100 [170]. The data represent two biological replicates with five technical reps (leaf from an individual plant) per treatment.

### Root length assays

Root length assays to quantify trehalose treatment effect on root length were performed as described with modification [8]. Tomato seeds were sterilized in 10% bleach for 10 min, then 95% ethanol for 10 min, then washed five times with sterile water and germinated in the dark on 1% water agar plates at room temperature. Four-day-old tomato seedlings were transferred to germination pouches (Mega International, Newport, MN) then watered with either 10 mL of water or 30 mM trehalose solution and incubated in a 28˚C growth chamber. Following the initial trehalose treatment, pouches were kept moist as needed with sterile water and imaged ten days post-treatment. This experiment was performed three times with ten tomato seedlings/pouch/treatment. New root growth was calculated as % new root growth = (new growth/ total root length) x 100.

### Chlorophyll determination

To measure the amount of chlorophyll in tomatoes, we used a SPAD-502 meter (Konica Minolta, Inc.) [171]. Readings were performed on the first true leaves of two-week-old Bonny Best plants treated with 50 mL of a 30 mM trehalose solution or water. The data represent two biological replicates with ten plants per treatment per replicate.

### Bacterial leaf spot of tomato inoculations

*Xanthomonas gardneri* culture conditions, infection, and colonization analysis were as described [172]. *X. gardneri* strains were cultured on nutrient agar plates (Millipore-Sigma) supplemented with nalidixic acid (50 μg/mL) at 28˚C. Inoculum was grown for 48 h in a shaking incubator at 28˚C in nutrient broth supplemented with nalidixic acid. The inoculum was adjusted to $OD_{600}$ = 0.3 and diluted 1:100 in 400 mL sterile water with 75 μL Silwet (Fisher Scientific). Four-week-old Bonny Best tomato plants that were treated with water or 50 mL of 30 mM trehalose solution 72 h previously were dip-inoculated for 30 s in the inoculum. Plants were incubated in high humidity conditions (85–90%) in plastic containers in the growth chamber for 48 h. After 48 h, plants were moved to low humidity conditions during the day (taken out of containers, 45–60%) and high humidity conditions at night on a 12 h cycle. Bacterial spot symptoms appeared 5–6 days post-inoculation. Leaf tissue from individual plants was sampled at 3, 6, and 9 days post-inoculation. Two 2 $cm^2$ portions of leaf were sampled from each plant, ground in a Powerlyzer (Qiagen) in water, and serially diluted and enumerated on nutrient agar + nalidixic acid. The data represent three biological replicates with 12 technical reps/treatment/timepoint.

### DAB staining tomato leaves for ROS production

Tomato leaves were stained with 3,3'-Diaminobenzidine (DAB, Sigma-Aldrich) as described [173]. Leaves from 4-5-week-old Bonny Best tomato plants were infused with water, 30 mM of trehalose solution, or $1x10^9$ CFU/mL *R. solanacearum* cells by pricking with a needle then flushing the apoplast with the solution. Plants were returned to the growth chamber overnight. At 24 h, leaves were cut and placed in 0.1 mg/mL DAB solution prepared at pH 3.8 at room temperature [174]. Containers holding the stain and leaves were wrapped in aluminum foil to

exclude light and incubated for 8 h at room temperature with gentle shaking. At 8 h, leaves were bleached by boiling in 95% ethanol for 10 min, then further cleared with overnight incubation in 70% ethanol before imaging. These experiments were performed three times with leaves from four individual tomato plants per treatment.

## Statistical analyses

Statistical analyses and graphs were generated with GraphPad Prism 8.

## Supporting information

**S1 Table. Strains and primers used in this study.**
(PDF)

**S2 Table. Concentration of hormones, oxylipins, and phenylpropanoids in tomato xylem sap following various treatments.**
(PDF)

**S3 Table. Metabolite key for hormone.**
(PDF)

**S1 Fig. ABA and DMSO are not toxic to *Rs*.** Growth of Rs cultured in rich CPG broth supplemented with ABA in DMSO, or water with DMSO because ABA stock solutions were dissolved in DMSO (100 μM treatments contained 0.1 μL of either DMSO or ABA stock solution, and 50 μM treatments contained 0.05 μL DMSO or ABA stock solution) (ANOVA Area Under Curve/AUC, Fisher's LSD multiple comparisons to CPG control, 100 μM DMSO, $P$ = .55; 50 μM DMSO, $P$ = .53; 100 μM ABA, $P$ = .67; 50 μM ABA, $P$ = .45). Growth was measured spectrophotometrically using a Bio-Tek plate reader; the data represent eight technical reps/treatment. The bars represent the standard error of the mean.
(PDF)

**S2 Fig. Trehalose treatment temporarily improves water efficiency without affecting photosynthesis; trehalose treatment does not affect root growth and relative water content of leaves but lowers chlorophyll concentration. A)** Water efficiency, measured with a LI-COR as photosynthetic rate/transpiration, in trehalose treated plants compared to water only controls (t-test, $P<0.5$). **B)** Photosynthesis, measured as $A_{net}$, in trehalose treated plants (Mixed model ANOVA, Tukey's HSD, $P>.05$). The water efficiency and photosynthetic data represent 20 plants/treatment. LI-COR measurements were taken ~10 min after treatment and then in the morning every time point after. Shaded bars on the graphs represent night periods. All data were calculated from the same LICOR time points. The bars on the data points represent the standard deviation. **C)** Relative water content of leaves was calculated from the fresh weight, hydrated weight, and dry weight of leaves from water control or trehalose treated BB tomatoes (t-test, $P$ = .14). The data represent two biological replicates with five technical replicates (leaves)/treatment. The bars represent the standard deviation. **D)** 'Bonny Best' tomato seedlings were placed in root growth pouches, watered with 10mL of either water or trehalose for one day, then watered as needed for ten days. Root length was measured after ten days (t-test, $P$ = .31). The data represent three biological replicates with ten roots/treatment. Percent of new root growth was calculated as %new root growth = (new growth/total root length)*100. The bars represent the standard deviation. **E)** SPAD meter measurements of chlorophyll content in leaves following trehalose treatment (2-way ANOVA Fishers LSD multiple comparison, * = $P<0.5$ or lower). The bars represent the standard deviation.
(PDF)

**S3 Fig. Trehalose treatment does not nutritionally enrich the soil environment for *R. solanacearum*, and its protection is limited to root application. A)** Disease progress curve of plants treated with trehalose or water, then infected with either wild-type *Rs* or a Δ*treA* mutant unable to catabolize trehalose (one-way ANOVA of areas under the curve, WT H$_2$0 vs. WT tre, *P* = .0008, Δ*treA* H$_2$0 vs. Δ*treA* tre, *P* = .017). The data represent three bioreps each containing 13–15 plants per treatment. The bars represent the standard error. **B)** Disease development in plants sprayed once with 30 mM trehalose or water, and then soil-soak inoculated with *Rs* 48 h later (ANOVA of AUC, Fisher's LSD multiple comparisons to H$_2$0 soil, H$_2$0 spray, *P* = .67; tre spray, *P* = .63; tre soil, *P* = .63). The data represent three biological replicates each containing fifteen plants per treatment. The bars represent the standard error. **C)** Colonization of tomato leaves by the bacterial leaf spot pathogen *Xanthomonas gardneri*; plants were dip-inoculated with *X. gardneri* 72 h after root treatment with water or 30 mM trehalose (Mann-Whitney, day 3, *P* = .06; day 6, *P* = .61; day 9, *P* = .08). The data represent three biological replicates with twelve plants/treatment/timepoint. The graph displays the geometric means; bars indicate the geometric standard deviation of the data.
(PDF)

**S4 Fig. Infusing leaves with trehalose did not trigger ROS production in 'Bonny Best' tomato leaves.** DAB staining of 'Bonny best' tomato leaves infused with water, 10$^9$ CFU/mL *Rs*, or 30 mM trehalose solution to assess the effect of trehalose treatment on ROS production. The data represent three biological replicates, with four plants per biological replicate per treatment. Photos are representative samples and images were uniformly sharpened 25% to increase contrast.
(PDF)

## Acknowledgments

The authors thank Dr. Raka Mitra (Carleton College) for manuscript review and valuable discussions. We are grateful to Dr. Eric Kruger (UW-Madison Dep't of Forest and Wildlife Ecology) for the use of equipment and Dr. Kim Cowles (UW-Madison Dep't of Plant Pathology) for *X. gardneri* cultures and technical advice.

## Author Contributions

**Conceptualization:** April M. MacIntyre, Anjali S. Iyer-Pascuzzi, Caitilyn Allen.

**Data curation:** April M. MacIntyre, Valerian Meline, Zachary Gorman.

**Formal analysis:** April M. MacIntyre, Valerian Meline.

**Funding acquisition:** Caitilyn Allen.

**Investigation:** April M. MacIntyre, Valerian Meline, Zachary Gorman, Steven P. Augustine, Carolyn J. Dye, Corri D. Hamilton, Caitilyn Allen.

**Methodology:** April M. MacIntyre, Steven P. Augustine, Corri D. Hamilton, Michael V. Kolomiets.

**Project administration:** Caitilyn Allen.

**Resources:** Anjali S. Iyer-Pascuzzi.

**Supervision:** Anjali S. Iyer-Pascuzzi, Michael V. Kolomiets, Katherine A. McCulloh, Caitilyn Allen.

**Writing – original draft:** April M. MacIntyre.

**Writing – review & editing:** Anjali S. Iyer-Pascuzzi, Michael V. Kolomiets, Katherine A. McCulloh, Caitilyn Allen.

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
