## [Decision Letter · Decision Letter 0]

9 Mar 2022

PONE-D-22-04449Trehalose increases tomato drought tolerance, induces defenses, and increases resistance to bacterial wilt diseasePLOS ONE

Dear Dr. Allen,

Thank you for submitting your manuscript to PLOS ONE. After careful consideration, we feel that it has merit but does not fully meet PLOS ONE’s publication criteria as it currently stands. Therefore, we invite you to submit a revised version of the manuscript that addresses the points raised during the review process.

The staff editors from PLOS ONE have some minor requirements for this to meet the journal publication criteria, please see the '**Journal Requirements**'.

We look forward to receiving your revised manuscript.

Kind regards,

Ching-Hong Yang

Academic Editor

PLOS ONE

**Journal Requirements**:

Please consider the comments of the reviewers regarding the labelling of figures, carefully proofread the manuscript and check for correct labelling/copyediting.

Additional Editor Comments (if provided):

Reviewers' comments:

Reviewer's Responses to Questions

**Comments to the Author**

1. Is the manuscript technically sound, and do the data support the conclusions?

Reviewer #1: Yes

Reviewer #2: Yes

2. Has the statistical analysis been performed appropriately and rigorously? 

Reviewer #1: Yes

Reviewer #2: Yes

3. Have the authors made all data underlying the findings in their manuscript fully available?

Reviewer #1: Yes

Reviewer #2: Yes

4. Is the manuscript presented in an intelligible fashion and written in standard English?

Reviewer #1: Yes

Reviewer #2: Yes

5. Review Comments to the Author

Reviewer #1: The manuscript titled “Trehalose increases tomato drought tolerance, induces defenses, and increases resistance to bacterial wilt disease” studies the influence of treating trehalose on tomato. The manuscript describes a comprehensive study on the relationship of trehalose, Ralstonia solanacearim, plant defenses and drought tolerance of tomato plants. The work is well structured and is presented clearly. Techniques and methodologies are suitable and well described. Authors presents results of original research and the rest of the requirements of the journal are fulfilled. I have just a few minor comments and suggestions for corrections:

Results

Line 261: Fig 3B should be corrected to Fig 3C.

Line 264: Fig 3C should be corrected to Fig 3D.

Supplemental Figure 3C, colonization of tomato leaves by the bacterial leaf spot pathogen Xanthomonas gardneri, …., day 9, p=08), the decimal point was missing.

Discussion

If possible, insert the corresponding figure numbers at the paragraph for easily reading. For example, page 16, line 464-465: In response to R. solanacearum infection, tomatoes downregulated multiple aquaporins, suggesting that plant may try to alter water flow during bacterial wilt (Fig 1).

Reviewer #2: The authors describe the role of trehalose presented in xylem sap of Ralstonia solanacearum infected-tomato, and confirm the link between plant responses to biotic and abiotic stress. All of these are consistent with their previous trehalose work in R. solanacearum. It is clear from this and previous works that plants perceive trehalose as a DAMP and respond with systemic disease resistance. In turn, R. solanacearum degrades trehalose as a counter-defense.

6. PLOS authors have the option to publish the peer review history of their article (what does this mean?). If published, this will include your full peer review and any attached files.

Reviewer #1: No

Reviewer #2: No

---

## [Author Response · Author response to Decision Letter 0]

14 Mar 2022

Responses to Reviews and Journal requirements

1.Please consider the comments of the reviewers regarding the labelling of figures, carefully proofread the manuscript and check for correct labelling/copyediting. 

-Thank you, copyediting has been completed and reviewer suggestions addressed.

-References have been reformatted to the Vancouver style. Species names have been italicized where needed (ref. 84, 85, 86). 

-Supplemental tables and figures now have their own reference list in the Vancouver style. 

- The citation list has been run through the Zotero/Retraction Watch creen and, to the best of our knowledge, none of these papers have been retracted. 

Comments to the Author

1. Is the manuscript technically sound, and do the data support the conclusions?

Reviewer #1: Yes

Reviewer #2: Yes

2. Has the statistical analysis been performed appropriately and rigorously?

Reviewer #1: Yes

Reviewer #2: Yes

3. Have the authors made all data underlying the findings in their manuscript fully available?

Reviewer #1: Yes

Reviewer #2: Yes

4. Is the manuscript presented in an intelligible fashion and written in standard English?

Reviewer #1: Yes

Reviewer #2: Yes

5. Review Comments to the Author

Reviewer #1: The manuscript titled “Trehalose increases tomato drought tolerance, induces defenses, and increases resistance to bacterial wilt disease” studies the influence of treating trehalose on tomato. The manuscript describes a comprehensive study on the relationship of trehalose, Ralstonia solanacearim, plant defenses and drought tolerance of tomato plants. The work is well structured and is presented clearly. Techniques and methodologies are suitable and well described. Authors present results of original research and the rest of the requirements of the journal are fulfilled. I have just a few minor comments and suggestions for corrections:

Results

Line 261: Fig 3B should be corrected to Fig 3C. -Thank you, this has been corrected.

Line 264: Fig 3C should be corrected to Fig 3D. -Thank you, this has been corrected.

Supplemental Figure 3C, colonization of tomato leaves by the bacterial leaf spot pathogen Xanthomonas gardneri, …., day 9, p=08), the decimal point was missing. --Thank you, this has been corrected.

Discussion

If possible, insert the corresponding figure numbers at the paragraph for easily reading. For example, page 16, line 464-465: In response to R. solanacearum infection, tomatoes downregulated multiple aquaporins, suggesting that plant may try to alter water flow during bacterial wilt (Fig 1). 

-Thank you for the suggestion. Figures asre now called out in the Discussion (see lines 464, 473, 459, 457, 488, 489, 498, 499, 509, 510-513, 527, 536, 550, 558, 565, 568, 578, 584-585, 588, 596, 599, 611, 618, and 639-640).

Reviewer #2: The authors describe the role of trehalose presented in xylem sap of Ralstonia solanacearum infected-tomato and confirm the link between plant responses to biotic and abiotic stress. All of these are consistent with their previous trehalose work in R. solanacearum. It is clear from this and previous works that plants perceive trehalose as a DAMP and respond with systemic disease resistance. In turn, R. solanacearum degrades trehalose as a counter-defense.

6. PLOS authors have the option to publish the peer review history of their article (what does this mean?). If published, this will include your full peer review and any attached files.

Do you want your identity to be public for this peer review? For information about this choice, including consent withdrawal, please see our Privacy Policy.

Reviewer #1: No

Reviewer #2: No

- Done, thank you.

---

## [Editor Report · Decision Letter 1]

17 Mar 2022

Trehalose increases tomato drought tolerance, induces defenses, and increases resistance to bacterial wilt disease

PONE-D-22-04449R1

Dear Dr. Allen,

We’re pleased to inform you that your manuscript has been judged scientifically suitable for publication and will be formally accepted for publication once it meets all outstanding technical requirements.

Kind regards,

Ching-Hong Yang

Academic Editor

PLOS ONE
---

## [Editor Report · Acceptance letter]

29 Mar 2022

PONE-D-22-04449R1 

Trehalose increases tomato drought tolerance, induces defenses, and increases resistance to bacterial wilt disease 

Dear Dr. Allen:

I'm pleased to inform you that your manuscript has been deemed suitable for publication in PLOS ONE. Congratulations! Your manuscript is now with our production department. 

Kind regards, 

on behalf of

Dr. Ching-Hong Yang 

Academic Editor

PLOS ONE